# EvoTest: Evolutionary Test-Time Learning for Self-Improving Agentic Systems

**Yufei He**[1]***Juncheng Liu**[2]**, Yue Liu**[1]**,Yibo Li**[1]**, Tri Cao**[1]**, Zhiyuan Hu**[1]†**Xinxing Xu**[2]‡**Bryan Hooi**[1]†

[1]National University of Singapore

[2]Microsoft Research

{yufei.he, yliu, liyibo, zhiyuan_hu}@u.nus.com,

tricao2001vn@gmail.com,

bhooi@comp.nus.edu.sg,

{juncheng.liu, xinxingxu}@microsoft.com

## Abstract

A fundamental limitation of current AI agents is their inability to learn complex skills on the fly at test time, often behaving like "clever but clueless interns" in novel environments. This severely limits their practical utility. To systematically measure and drive progress on this challenge, we first introduce the **Jericho Test-Time Learning (J-TTL)** benchmark. J-TTL is a new evaluation setup where an agent must play the same game for several consecutive episodes, attempting to improve its performance from one episode to the next. On J-TTL, we find that existing adaptation methods like reflection, memory, or reinforcement learning struggle. To address the challenges posed by our benchmark, we present **EvoTest**[1], an evolutionary test-time learning framework that improves an agent without any fine-tuning or gradients—by evolving the entire agentic system after every episode. EvoTest has two roles: the **Actor Agent**, which plays the game, and the **Evolver Agent**, which analyzes the episode transcript to propose a revised configuration for the next run. This configuration rewrites the prompt, updates memory by logging effective state–action choices, tunes hyperparameters, and learns the tool-use routines. On our J-TTL benchmark, EvoTest consistently increases performance, outperforming not only reflection and memory-only baselines but also more complex online fine-tuning methods. Notably, our method is the only one capable of winning two games (Detective and Library), while all baselines fail to win any.

## 1 Introduction

The pursuit of truly autonomous agents hinges on a critical human capability: the ability to learn "on the fly" (Maes, 1993; Franklin & Graesser, 1996). When faced with a new task, humans can attempt it, reflect on their successes and failures, formulate a better strategy, and try again. By contrast, most AI agents arrive at deployment with a fixed policy, behaving like "clever but clueless interns" that can execute instructions but cannot reform their own process from experience (Huang et al., 2024; Talebirad & Nadiri, 2023; Wang et al., 2024; 2025; Hou et al., 2023). This gap severely limits their reliability in dynamic settings. While the field acknowledges this problem, progress has been hampered by a lack of standardized testbeds designed specifically to measure an agent's capacity for rapid, in-session improvement (Zhou et al., 2023; Mialon et al., 2023; He et al., 2025a; 2024; 2026; Li et al., 2026; Yang et al., 2026; Sui et al., 2024a).

To address this, we first introduce the **Jericho Test-Time Learning (J-TTL)** benchmark, a new evaluation framework designed to systematically measure and drive progress in on-the-fly agent learning. The benchmark's core task is straightforward: an agent must play the same complex, text-based adventure game (Hausknecht et al., 2020) for a series of consecutive attempts ("episodes"). In

---

* The work was done when the author was an intern at Microsoft Research Asia - Singapore

† Corresponding author.

[1]The code is available at https://github.com/yf-he/EvoTest

each episode, the agent interacts with the environment through a standard loop: it receives a textual observation of its surroundings (state), submits a natural-language command (action), and receives a numerical score change (reward). These games are difficult for LLM agents because they feature complex puzzles, long-range planning, sparse rewards (many critical actions yield no points), and irreversible consequences (a single wrong move can make the game unwinnable). The agent's goal is structured at two levels: 1) The *Episodic Goal*: Maximize the final score within a single playthrough. 2) The *Learning Goal*: Play the same game repeatedly and progressively increase its final score from one episode to the next, using only the experience gathered within that single session.

The J-TTL benchmark starkly reveals the inadequacies of existing adaptation paradigms. Consider a simple but critical failure in the game *Detective*: an agent gets stuck in a navigation loop by repeatedly attempting an invalid action, such as `GO WEST`, which the game rejects with `"You can't go that way."` This seemingly simple failure reveals deep flaws in current adaptation methods: A **Static** agent has no learning mechanism and will likely repeat this error in every episode, leading to a flat, low-scoring performance. An **SFT (online)** agent will have no good data to learn from in this failed episode. It is trapped because it cannot generate the very data it needs to improve. An **Reinforcement Learning (RL)(online)** agent receives a `reward=0` for the invalid move, which is a weak signal in a sparse-reward environment. A single update based on this noisy signal is insufficient to correct the policy, demonstrating a failure of credit assignment. Methods based on **reflection**, such as Reflexion (Shinn et al., 2023), modify the agent's prompt with summaries of past failures. While useful, it does not alter the agent's core decision-making logic or its use of tools. Similarly, advanced **memory systems** (Packer et al., 2023; Zhong et al., 2024) improve an agent's ability to recall information but do not teach it how to act differently. On the other end of the spectrum, RL and online fine-tuning are fundamentally ill-suited for the test-time learning setting. These methods are too slow and data-inefficient for the rapid learning J-TTL demands. To meet the challenge posed by our benchmark, we introduce **EvoTest, an evolutionary test-time learning framework designed for rapid, holistic adaptation without fine-tuning.** EvoTest decouples acting from adaptation using two distinct roles: an **Actor Agent** that plays a full episode and an **Evolver Agent** that improves the system between independent episodes. After each episode, the Evolver Agent analyzes the full transcript and proposes a revised configuration for the entire agentic system. This process of whole-system evolution involves:

1. Rewriting the guiding prompt to encode new strategies;
2. Updating a structured deployment-time memory with records of successful and failure actions;
3. Tuning decision-making hyperparameters like temperature and exploration strength;
4. Refining the tool-use routines that govern how and when memory or python code is accessed.

By evolving the agent configuration, EvoTest transforms the narrative of one episode into multifaceted improvements for the next attempt, enabling a deeper form of learning than prior methods. We summarize our contributions as follows:

- **A Benchmark for Test-Time Learning:** We propose J-TTL, a benchmark using Jericho games to measure an agent's on-the-fly learning ability across a series of playthroughs of the same game.
- **A Test-time Learning Algorithm:** We propose EvoTest, an evolutionary agent learning framework that evolves the entire agentic system (policy, memory, tool-use routines, and hyperparameters) via transcript-level analysis without gradients or fine-tuning.
- **State-of-the-Art Empirical Results:** We demonstrate on the J-TTL benchmark that EvoTest shows a 38% improvement over the strongest prompt-evolution baseline and a 57% improvement over online RL, outperforming all strong reflection-based, memory-based, and gradient-based baselines on every game.

## 2 RELATED WORK

**From Static Agents to Test-Time Learning.** The majority of current AI agents, while capable, operate with static configurations that are manually designed and fixed after deployment (Wang et al., 2024; Xi et al., 2025; He et al., 2025b; Chen et al., 2025b;a; He et al., 2025c;d; Chen et al., 2025c). This limits their ability to adapt to novel situations, a key challenge motivating the development of "self-improving AI agents" (Gao et al., 2025b; Fang et al., 2025; Gao et al., 2025a; Sui et al., 2025;

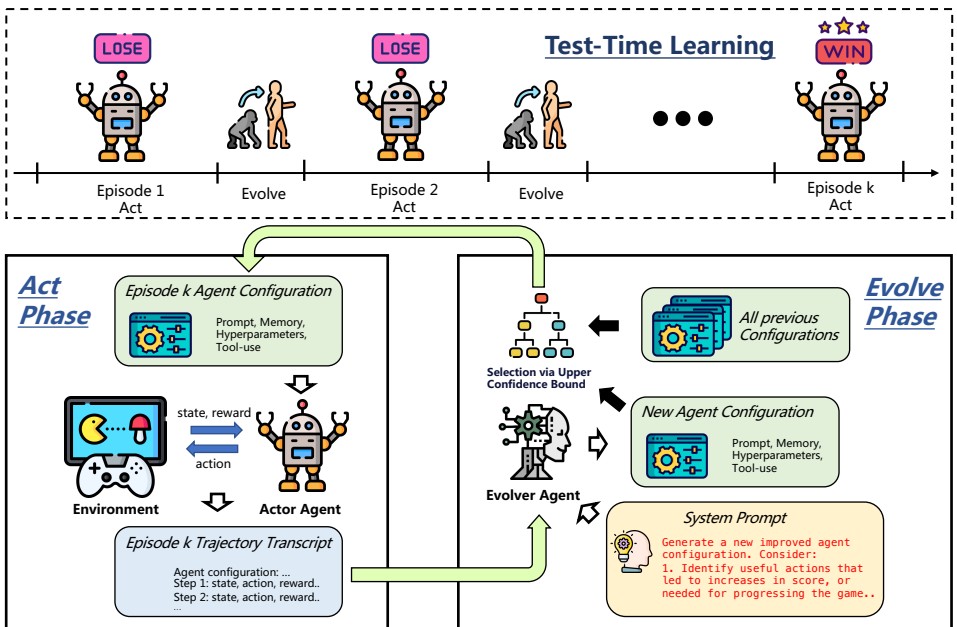

Figure 1: The `EvoTest` architecture, designed to enable **test-time learning** (TTL). The agent operates in a continuous Act-Evolve loop across multiple attempts at the same task. After each episode, the Evolver Agent analyzes the full trajectory transcript—rich narrative feedback to perform gradient-free, **whole-system evolution** on the agent's entire configuration. This allows the agentic system to self-improve on the fly, directly from its own experience at test time.

2024b; Liu et al., 2025b). A prominent line of work enables agents to learn from past mistakes without updating weights. Reflexion (Shinn et al., 2023), a key baseline for our work, allows an agent to verbally reflect on trajectory failures and append these reflections to its prompt for subsequent episodes. Other approaches focus on enhancing agent memory. For instance, MemGPT (Packer et al., 2023) provides agents with a structured memory system to manage long contexts. Beyond reflection/memory, Uncertainty of Thoughts (Hu et al., 2024) adds test-time *uncertainty-aware planning*, deciding when to ask, verify, or revise without weight updates. While MemoryBank (Zhong et al., 2024) uses hierarchical summarization to retain information over long interactions.

**Self-Evolving Agentic Systems.** Another active area of research is the automated optimization of prompts that guide agent behavior (Liu et al., 2025a; Zhu et al., 2026; Hu et al., 2026). Generative approaches like APE (Zhou et al., 2022) and OPRO (Yang et al., 2023) use a powerful LLM to propose and score new prompts, iteratively refining them based on performance. Gradient-inspired methods like TextGrad (Yuksekgonul et al., 2024) refine prompts using LLM-generated textual feedback. Closely related to our work are evolutionary methods such as AlphaEvolve (Novikov et al., 2025), Promptbreeder (Fernando et al., 2023), and EvoPrompt (Guo et al., 2024), which maintain a population of prompts and apply genetic operators like mutation and crossover to discover more effective instructions. EvoTest generalizes prompt evolution to whole-system evolution, optimizing the entire agentic configuration—including the prompt, memory, hyperparameters, and tool-use routines. This allows for more holistic adaptations, such as tuning exploration strength, that are beyond the scope of prompt-editing alone. This vision for unified optimization is shared by EvoAgent (Yuan et al., 2024) and MASS (Zhou et al., 2025); Beyond 'Aha!' (Hu et al., 2025) complements this by aligning meta-abilities rather than only task prompts or single components.

## 3 THE JERICHO TEST-TIME LEARNING (J-TTL) BENCHMARK

To systematically measure and drive progress in on-the-fly agent learning, we introduce the **Jericho Test-Time Learning (J-TTL)** benchmark. This benchmark is built upon the Jericho (Hausknecht et al., 2020)[1] suite of Interactive Fiction (IF) games. IF games are fully text-based simulation en-

---

[1]Jericho is available at `https://github.com/Microsoft/jericho`

vironments where an agent issues text commands to effect change in the environment and progress through a story. While the richness of these environments makes them a challenging testbed for AI, existing evaluation has primarily focused on single-episode performance or generalization across different games (Hausknecht et al., 2020; Gulcehre et al., 2020; Li et al., 2025). The J-TTL benchmark refocuses the evaluation on a different, critical axis: an agent's ability to learn and improve its strategy through repeated attempts at the same complex task within a single test session.

**Datasets.** We use publicly available Jericho games that vary in difficulty and puzzle structure, including *Detective*, *Library*, *Zork1*, *Zork3*, *Balances*, and *Temple*. Games are launched via Jericho with default scoring. Each episode is capped by a step limit ($T = 110$ unless stated otherwise).

**The Jericho Game.** We model a Jericho game (Hausknecht et al., 2020) as a Partially Observable Markov Decision Process (POMDP), defined by the tuple $(\mathcal{S}, \mathcal{A}, \mathcal{T}, \mathcal{R}, \Omega, T)$. Here, $\mathcal{S}$ is the latent state space, and $\mathcal{A}$ is the (infinite) combinatorial action space of natural language commands. At each step $t$, an agent in a latent state $s_t \in \mathcal{S}$ takes an action $a_t \in \mathcal{A}$, causing a transition to a new state $s_{t+1} \sim \mathcal{T}(\cdot \mid s_t, a_t)$ and yielding a scalar reward $r_t = \mathcal{R}(s_t, a_t)$. The agent does not observe the true state $s_t$ but instead receives a textual observation $o_t \sim \Omega(\cdot \mid s_t)$. An episode is a trajectory of interactions with a finite horizon of $T$ steps:

$$\tau^{(e)} \triangleq \left( o_1^{(e)}, a_1^{(e)}, r_1^{(e)}, \ldots, o_T^{(e)}, a_T^{(e)}, r_T^{(e)} \right). \tag{1}$$

The total return for an episode $e$ is the sum of its rewards, $R(e) \triangleq \sum_{t=1}^{T} r_t^{(e)}$.

**Test-Time Learning Process on Jericho.** The J-TTL benchmark protocol consists of a **session** of $K$ consecutive **episodes** played in a single game. After each episode concludes, the game environment is reset to its identical initial state, $s_0^{(e)} = s_{\text{init}}$ for all $e \in \{1, \ldots, K\}$. This ensures that any performance improvement is solely attributable to the agent's internal learning process.

Let the agent be parameterized by a set of learnable components $\theta \in \Theta$. In episode $e$, the agent's behavior is governed by a policy conditioned on its current learnable components, $\pi_{\theta^{(e)}}$.

The core of test-time learning lies in the update rule an agent applies between episodes. After completing episode $e$ and collecting the trajectory data $\tau^{(e)}$, the agent updates its learnable components for the next episode:

$$\theta^{(e+1)} = U(\theta^{(e)}, \tau^{(e)}), \tag{2}$$

where $U : \Theta \times \mathcal{T}_{\text{hist}} \to \Theta$ is the agent's learning algorithm.

**Evaluation Metrics.** For a test-time learning session consisting of $K$ episodes, we record total score for each episode, $R(e)$. This yields a performance sequence for the entire session:

$$\{R(1), R(2), \ldots, R(K)\}. \tag{3}$$

From this sequence, we derive two metrics:

- **Learning Curve.** A plot of the final return $R(e)$ against the episode index $e$. This curve provides a direct visualization of an agent's learning progress.
- **Area Under the Curve (AUC).** For quantitative comparison, we define the AUC as a normalized ratio. It measures the agent's total achieved score against the maximum possible score over the session:

$$\text{AUC} = \frac{\sum_{e=1}^{K} R(e)}{K \cdot R_{\max}}, \tag{4}$$

where $R_{\max}$ is the maximum achievable score in a single episode of the game. This metric yields a value between 0 and 1, facilitating comparison across games with different scoring scales.

## 4    THE EVOTEST FRAMEWORK

To address the challenges posed by the J-TTL benchmark, we introduce **EvoTest**, an evolutionary test-time learning framework, as illustrated in Figure 1. Unlike methods that perform gradient-based updates on model weights, EvoTest operates on a fixed, non-trainable backbone LLM. It achieves learning by evolving the agent's entire high-level configuration between episodes, leveraging the rich, narrative feedback from game transcripts rather than just sparse, scalar rewards.

## 4.1 THE AGENTIC CONFIGURATION

In the context of EvoTest, the general learnable components $\theta$ from the J-TTL formulation (Section 3) are instantiated as a holistic **agentic configuration**, denoted by $\chi \in \mathcal{X}$. This configuration is a tuple $\chi = (p, M, h, u)$ that defines the agent's complete operational strategy:

- **Policy Prompt ($p$):** A system prompt that provides high-level strategic guidance, heuristics, and behavioral guardrails to the backbone LLM.
- **Deployment-time Memory ($M$):** A structured, queryable database populated by the Evolver Agent after each episode. It stores the agent's prior experiences, effectively creating a persistent knowledge base. The memory is organized into distinct components, such as: (a) a *success memory* that logs state-action pairs which led to score increases (e.g., `(state_hash, action) -> score_delta`), and (b) a *failure memory* that records patterns associated with stalls or negative outcomes (e.g., repetitive action loops in a specific location). This allows the agent to recall specific, effective actions and avoid known pitfalls from past playthroughs. For a detailed breakdown of the memory's data structure and concrete examples of its contents, please see Appendix J.
- **Hyperparameters ($h$):** A set of parameters controlling the LLM's inference and the agent's decision-making, such as temperature, exploration strength, and stopping criteria.
- **Tool-Use Routines ($u$):** Active components executed at each decision-making step to operationalize knowledge and create useful state abstractions. These routines consist of two functions:
  - **Memory Interaction Logic:** A set of rules governing how to query the memory ($M$). Before deciding on an action, this routine might check if the current game state exists in memory. If a match is found in the success memory, the routine can inject the proven action into the LLM's prompt as a strong suggestion (e.g., "Hint: In this exact situation before, the action 'unlock door with key' worked well.").
  - **State Abstraction Logic:** An evolvable Python function—the *state extractor*—that processes the raw, verbose game history into a concise summary of progress. Instead of forcing the LLM to re-read the entire history, this tool parses the log for key environmental cues (e.g., the text `"The ancient scroll disintegrates, revealing a map."`) and returns a short, meaningful milestone string (e.g., `"Milestone:  Found the map."`). This abstracted state is included in the prompt at every step, providing efficient situational awareness.

The agent's policy in episode $e$, $\pi_{\chi^{(e)}}$, is therefore a function of the fixed LLM's behavior as modulated by this entire configuration.

## 4.2 THE TWO-AGENT LEARNING LOOP

EvoTest operationalizes the test-time learning update rule, $\chi^{(e+1)} = U(\chi^{(e)}, \tau^{(e)})$, through a cooperative two-agent design that separates acting from adaptation.

**1. The Actor Agent.** For a given episode $e$, the Actor Agent is provided with a single, fixed configuration $\chi^{(e)}$. It plays through the episode by repeatedly querying the backbone LLM, conditioning its actions on the current observation $o_t$ and any information retrieved from its memory $M^{(e)}$ as dictated by its tool-use routines $u^{(e)}$. The result of its playthrough is the trajectory $\tau^{(e)}$ and the final return $R(e)$.

**2. The Evolver Agent.** After the episode, the Evolver Agent takes the trajectory transcript $\tau^{(e)}$ and the parent configuration $\chi^{(e)}$ as input. It performs **whole-system evolution** by generating a set of new candidate child configurations, $C^{(e+1)} = \{\tilde{\chi}_1^{(e+1)}, \ldots, \tilde{\chi}_m^{(e+1)}\}$. This is the core learning step, where the Evolver uses an LLM to analyze the previous episode's trajectory and propose improvements to the agentic configuration. The evolutionary operators include:

- **Prompt Mutation:** The Evolver rewrites the policy prompt $p^{(e)}$ to create a new prompt $\tilde{p}$. It incorporates strategies that proved effective (e.g., adding "always examine objects before taking them") or adds explicit rules to prevent observed failures (e.g., "avoid repetitive navigation loops").
- **Memory Update:** The Evolver programmatically parses the transcript $\tau^{(e)}$ to identify and log important events. It records state-action pairs $(o_t, a_t)$ that immediately preceded a score increase in a

"success" table. It also identifies sequences of interactions that led to no progress and records them in a "failure" table. This updated memory $M^{(e+1)}$ is then inherited by all child configurations.

- **Hyperparameter Tuning:** The Evolver proposes adjustments to the hyperparameters $h^{(e)}$ to create $\tilde{h}$. For example, if the transcript shows the agent was stuck in a repetitive loop, the Evolver might suggest increasing the LLM's 'temperature' to encourage more diverse actions.
- **Tool-Use Refinement:** The Evolver can modify the logic in $u^{(e)}$ to create $\tilde{u}$, changing when or how the agent consults its memory. For example, it might strengthen a rule in the prompt from a general suggestion to a direct instruction, such as telling the agent it must check its success memory before every action. The specification of these tools and the required JSON output format are detailed in the master prompt provided in Appendix H.

A child configuration $\tilde{\chi}$ is a new combination of these potentially mutated components, representing a new hypothesis for a more effective agent.

### 4.3 CONFIGURATION SELECTION VIA UCB

After the Evolver generates a set of new "child" configurations, EvoTest decide which one to use for the next episode. This creates a classic dilemma: should we *exploit* a configuration that has worked well in the past, or should we *explore* a new, untested one that might be even better?

To manage this trade-off, we select the next configuration, $\chi^{(e+1)}$, from a candidate pool containing the parent from the previous episode, $\chi^{(e)}$, and its new children, $C^{(e+1)}$. The selection is guided by the Upper Confidence Bound (UCB) algorithm, a strategy from multi-armed bandit theory (Vermorel & Mohri, 2005) designed to manage the exploration-exploitation dilemma. The rule selects the configuration that maximizes the following score:

$$\chi^{(e+1)} = \underset{\tilde{\chi} \in \{\chi^{(e)}\} \cup C^{(e+1)}}{\arg\max} \left( \hat{\mu}(\tilde{\chi}) + \beta \sqrt{\frac{\log N}{1 + n(\tilde{\chi})}} \right) \tag{5}$$

The score for each configuration $\tilde{\chi}$ is a sum of two parts:

- **Performance Term ($\hat{\mu}(\tilde{\chi})$):** This is simply the average score the configuration has achieved so far. It encourages us to re-use configurations that have a good track record.
- **Exploration Bonus ($\beta\sqrt{\cdots}$):** This term gives a boost to configurations that are less certain about. $n(\tilde{\chi})$ is the number of times a configuration has been tried, $N$ is the total number of episodes completed, and $\beta$ is a hyperparameter controlling the exploration strength. The bonus is higher for configurations that have been tried fewer times ($n(\tilde{\chi})$ is small), making novel mutations attractive.

At the start of each new episode, the UCB algorithm selects a single configuration, which is then used for the entire duration of that episode. The episode's final score is then used to update the statistics for that chosen configuration. Crucially, this UCB approach also makes our learning process more stable and helps prevent sharp drops in performance. A simpler, greedy strategy might get fooled by a new configuration that gets a lucky high score and then repeatedly fails. UCB avoids this pitfall. Because the reliable parent configuration is always in the selection pool, if a new child performs poorly after its initial trial, its average score ($\hat{\mu}$) will drop. The UCB rule can then naturally "fall back" to the time-tested parent, which retains its high score. This acts as a safety net, preventing the system from getting stuck on a bad evolutionary path and ensuring a more consistent improvement.

## 5 EXPERIMENTS

Our experiments are designed to answer three central research questions regarding test-time learning on our J-TTL benchmark:

- **RQ1:** Does test-time learning (TTL) lead to meaningful performance improvements on complex tasks compared to a non-learning agent?
- **RQ2:** Does our proposed EvoTest framework enable more effective test-time learning compared to existing methods, such as those based on memory, reflection, and prompt optimization?

Table 1: Comparison of Area Under the Curve (AUC) scores on the J-TTL benchmark across six Jericho games for two backbone LLMs: `google/gemini-2.5-flash` (*G*) and `anthropic/claude-4-sonnet-20250522` (*C*). Weight-update methods use a separate backbone and are not distinguished by model. Higher values indicate better overall performance throughout the test-time learning session. The best performance in each column is highlighted in **bold**. Our method, **EvoTest**, consistently outperforms all baselines across both backbones.

| Method | Detective G | Detective C | Library G | Library C | Zork1 G | Zork1 C | Zork3 G | Zork3 C | Balances G | Balances C | Temple G | Temple C | Avg. G | Avg. C |
|---|---|---|---|---|---|---|---|---|---|---|---|---|---|---|
| ***Non-learning Baseline*** | | | | | | | | | | | | | | |
| Static | 0.21 | 0.23 | 0.15 | 0.16 | 0.03 | 0.04 | 0.05 | 0.06 | 0.11 | 0.12 | 0.08 | 0.09 | 0.11 | 0.12 |
| ***Memory-based & Reflection-based Methods*** | | | | | | | | | | | | | | |
| Memory | 0.25 | 0.26 | 0.18 | 0.20 | 0.04 | 0.05 | 0.06 | 0.07 | 0.13 | 0.14 | 0.10 | 0.11 | 0.13 | 0.14 |
| RAG | 0.32 | 0.34 | 0.24 | 0.25 | 0.07 | 0.08 | 0.09 | 0.10 | 0.18 | 0.20 | 0.15 | 0.16 | 0.18 | 0.19 |
| Summary | 0.45 | 0.47 | 0.33 | 0.35 | 0.12 | 0.13 | 0.15 | 0.16 | 0.25 | 0.27 | 0.21 | 0.22 | 0.25 | 0.27 |
| Reflexion | 0.58 | 0.60 | 0.41 | 0.44 | 0.09 | 0.11 | 0.25 | 0.27 | 0.30 | 0.32 | 0.29 | 0.31 | 0.32 | 0.34 |
| ***Automated Prompt Optimization Methods*** | | | | | | | | | | | | | | |
| TextGrad | 0.61 | 0.62 | 0.45 | 0.47 | 0.11 | 0.13 | 0.28 | 0.30 | 0.16 | 0.18 | 0.23 | 0.25 | 0.31 | 0.33 |
| Promptbreeder | 0.63 | 0.65 | 0.47 | 0.49 | 0.10 | 0.12 | 0.29 | 0.31 | 0.23 | 0.25 | 0.30 | 0.32 | 0.34 | 0.36 |
| EvoPrompt | 0.65 | 0.67 | 0.48 | 0.50 | 0.10 | 0.12 | 0.30 | 0.32 | 0.24 | 0.26 | 0.27 | 0.29 | 0.34 | 0.36 |
| ***Weight-Update Methods (Online Fine-Tuning)*** | | | | | | | | | | | | | | |
| SFT (online) | 0.40 | | 0.30 | | 0.11 | | 0.18 | | 0.22 | | 0.19 | | 0.23 | |
| GRPO (online) | 0.55 | | 0.38 | | 0.07 | | 0.22 | | 0.30 | | 0.26 | | 0.30 | |
| **EvoTest (Ours)** | **0.94** | **0.95** | **0.77** | **0.80** | **0.14** | **0.16** | **0.35** | **0.38** | **0.32** | **0.35** | **0.31** | **0.34** | **0.47** | **0.50** |

- **RQ3:** How does the gradient-free, evolutionary approach of EvoTest compare to traditional gradient-based RL methods in the context of test-time learning?

## 5.1 SETUP

**Backbone LLM.** We use two powerful API models for the Actor Agent: the cost-effective `google/gemini-2.5-flash` and the highly capable `anthropic/claude-4-sonnet-20250522`. The Evolver Agent, which performs the most complex reasoning, is powered by `openai/o3-2025-04-16`. The fine-tuning baselines (SFT and GRPO) are implemented on `qwen/qwen3-32b`, a large open-source model. Experiments with more LLM backbones can be found in Appendix N

**Baselines.** As a foundational reference, a non-learning **Static** agent with a fixed configuration establishes zero-shot performance. The learning baselines are grouped into four main categories: (1) **Memory-based Methods**, which include (1a) **Memory**, which places the complete session transcript history into the context for in-context learning, automatically truncating the oldest parts if the context limit is exceeded, and (1b) **RAG**, which retrieves relevant snippets from past trajectories; (2) **Reflection-based Methods**, which learn by appending textual feedback to the prompt, including (2a) **Summary**, which uses an LLM to progressively summarize the entire history of all past transcripts and feeds this condensed summary into the context, and (2b) **Reflexion** (Shinn et al., 2023), which generates structured textual self-reflections after each episode to critique performance and guide the next attempt; (3) **Automated Prompt Optimization Methods**, which iteratively refine the guiding prompt, including (3a) **TextGrad** (Yuksekgonul et al., 2024), where after each episode, an optimizer LLM analyzes the trajectory to generate a "textual gradient"—a critique describing how to improve the prompt—which is then applied for refinement, and two evolutionary approaches, (3b) **Promptbreeder** (Fernando et al., 2023) and (3c) **EvoPrompt** (Guo et al., 2024), which evolve a population of prompts; and (4) **Weight-Update Methods**, which contrast with our gradient-free approach by performing online fine-tuning, including (4a) **SFT (online)**, which performs Supervised Fine-Tuning on the actor model after each episode using the state-action pairs collected from the trajectory, and (4b) **GRPO (online)** (Shao et al., 2024), which applies gradient-based Reinforcement Learning policy updates to the model using the scalar rewards collected during the episode. All methods are evaluated under the same step budget and use the same backbone LLMs for their respective roles to ensure a fair comparison. The detailed introduction can be found in appendix C, D.

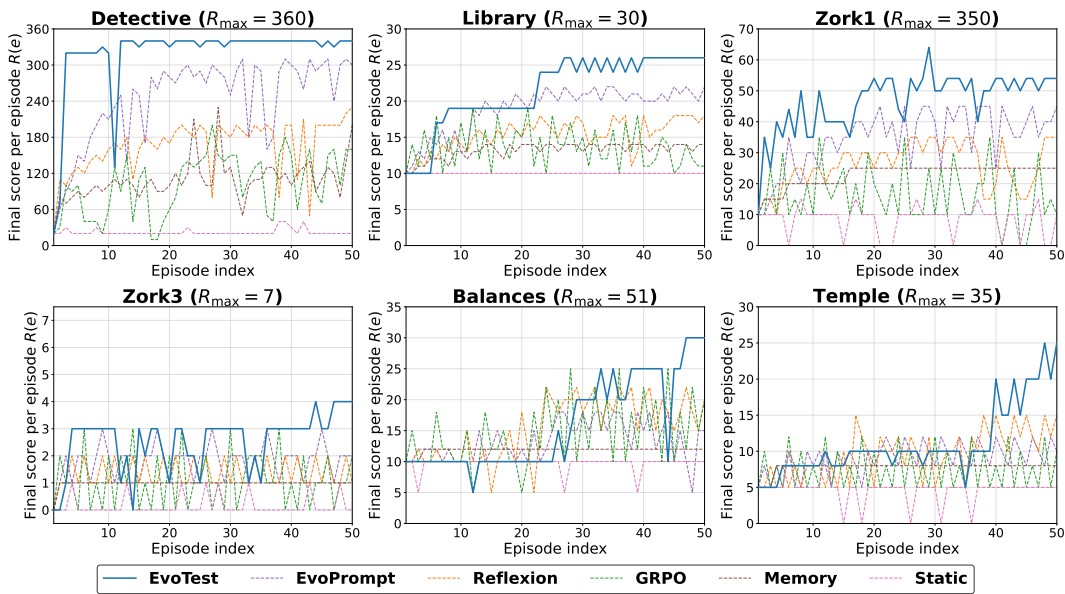

Figure 2: Learning curves showing final score per episode across six Jericho games with `google/gemini-2.5-flash` as LLM. EvoTest consistently demonstrates a steeper and more stable learning trajectory compared to baselines, validating its effectiveness for test-time learning.

## 5.2 RESULTS

Table 1 presents the AUC scores for all methods across the six selected Jericho games, while the learning curves in Figure 2 illustrate the per-episode performance progression. We analyze these results through the lens of our three research questions.

**RQ1: Test-Time Learning is Effective.** The results provide a clear affirmative answer. Across all games, every learning-based method achieves a higher average AUC score than the **Static** baseline (Table 1). The learning curves (Figure 2) further confirm this, showing that methods capable of learning from experience consistently exhibit upward-trending scores, whereas the Static agent's performance remains flat. This demonstrates that even with just a few attempts, test-time learning is a valid and effective paradigm for improving agent performance on complex, long-horizon tasks.

**RQ2: EvoTest Outperforms Existing Adaptation Methods.** EvoTest consistently and substantially outperforms all other gradient-free baselines. In Table 1, EvoTest achieves the highest AUC score on all six games, with an average score of 0.47/0.50—a significant improvement over the next best baseline, EvoPrompt (0.34/0.36). The learning curves reveal not just higher final scores but a steeper rate of improvement, indicating more efficient learning.

The key insight lies in the limitation of single-channel adaptation. **Memory** and **RAG** provide raw information but offer no strategic guidance, leading to a low performance ceiling. **Reflexion** and other prompt-focused optimizers like **Promptbreeder** perform better by refining strategy, but they are constrained to a single axis: the prompt. An agent can have a perfect prompt but still fail due to poor exploration (e.g., low temperature) or inefficient use of its knowledge. EvoTest's strength is its **holistic, whole-system evolution**. By concurrently optimizing the prompt, memory-access routines, and decision hyperparameters, it discovers and resolves complex performance bottlenecks that single-channel adaptations cannot. For example, it can learn to increase exploration temperature in early episodes and simultaneously add a new strategic heuristic to its prompt based on its findings, a multi-faceted adaptation that other methods are incapable of, as detailed and plotted in Appendix M.

**RQ3: Evolutionary Adaptation is More Data-Efficient than RL at Test Time.** The comparison between EvoTest and the weight-update methods clearly favors the evolutionary approach in this setting. EvoTest's average AUC (0.47/0.50) is substantially higher than that of **GRPO (online)** (0.30), the gradient-based RL baseline. This improvement indicates that EvoTest successfully addresses a fundamental challenge in test-time learning: extreme data scarcity.

Table 2: Practical costs for a single learning update.

| Method | Update Time | LLM Calls |
|---|---|---|
| SFT (online) | 5–10 min | 0 |
| GRPO (online) | 5–10 min | 0 |
| TextGrad | 30–50 sec | 2 |
| **EvoTest (Ours)** | **20–30 sec** | **1** |
| EvoPrompt | 20–30 sec | 1 |
| Reflexion | 15–25 sec | 1 |
| RAG | 5–15 sec | 0 (emb.) |
| Memory | <1 sec | 0 |

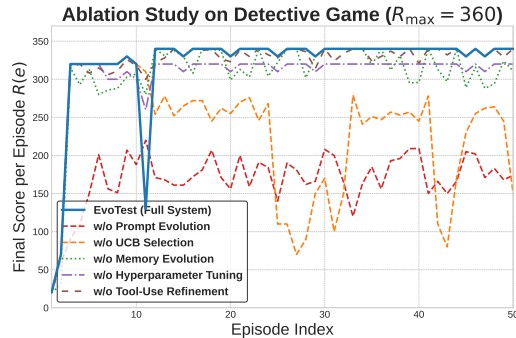

Figure 3: Learning curves from the component ablation study on the *Detective* game.

Table 3: Ablation study on EvoTest components, showing Area Under the Curve (AUC) scores.

| | *Detective* | *Zork1* | *Balances* |
|---|---|---|---|
| **EvoTest** | **0.94** | **0.14** | **0.32** |
| w/o Prompt | 0.52 | 0.05 | 0.16 |
| w/o UCB | 0.68 | 0.08 | 0.22 |
| w/o Memory | 0.82 | 0.11 | 0.28 |
| w/o Hyperpara. | 0.89 | 0.12 | 0.30 |
| w/o Tool-Use | 0.91 | 0.13 | 0.30 |

Table 4: Ablation on the Evolver agent's LLM. Performance, measured by AUC, correlates with model quality.

| Evolver LLM | *Detective* | *Zork1* | *Balances* |
|---|---|---|---|
| openai/o3 | **0.94** | **0.14** | **0.32** |
| deepseek/r1 | 0.90 | 0.12 | 0.29 |
| qwen3-32b | 0.82 | 0.10 | 0.25 |
| qwen3-8b | 0.68 | 0.07 | 0.20 |
| *Static* | *0.21* | *0.03* | *0.11* |

Moreover, EvoTest alleviate traditional RL's reliance on scalar rewards for credit assignment. In sparse-reward environments like Jericho, a single episode provides a noisy and insufficient signal for effective gradient updates. It is an inefficient way to learn from one complex success or failure. In contrast, EvoTest bypasses this issue by leveraging the **entire episode transcript as a rich, narrative feedback signal**. The Evolver Agent performs credit assignment through semantic analysis of the game's story, identifying causal chains of failure (e.g., "the agent got stuck in a loop here") and success. This allows it to make explicit, targeted, and structural edits to the agent's configuration. In essence, EvoTest shifts from credit assignment via backpropagation to **credit assignment via narrative analysis**, a more data-efficient mechanism for learning from a single experience.

## 5.3 MODEL ANALYSIS

**Ablation on Key Components.** Our ablation study reveals the distinct contributions of each component in the EvoTest framework. The AUC scores in Table 3 quantify the overall impact, showing that removing any component degrades performance. The largest performance drop occurs when removing prompt evolution (*w/o Prompt*), confirming that evolving the high-level policy is the primary driver of strategic adaptation.

The learning curves in Figure 3 offer deeper insight into the *dynamics* of these failures, particularly for the UCB ablation. While removing UCB also causes a significant drop in AUC, the curve reveals a more nuanced issue: instability. The *w/o UCB* agent, which uses a greedy selection strategy, is prone to catastrophic drops in performance. This happens when the agent over-commits to a high-risk mutation that achieved a lucky high score in one episode but is not robust. Without UCB's exploration mechanism—which encourages revisiting more reliable past configurations—the agent gets stuck on a suboptimal evolutionary path and is unable to correct its course after a poor choice. In contrast, the full EvoTest model leverages UCB to maintain a stable learning trajectory.

**Efficiency Analysis.** For learning at test-time, the update step between episodes is a major bottleneck. Our experiments show a clear divide in practicality between different approaches, as detailed in Table 2. Weight-update methods like online RL are not practical for this setting. A fine-tuning pass on one episode's data took 5 to 10 minutes on 4 H100 GPUs. This is not just slow; it demands expensive hardware, making it a non-starter for a system that needs to learn at test time. In contrast, EvoTest and other gradient-free methods operate on a different timescale. Instead of a costly training

Table 5: Ablation on the structure of prompt evolution. "Simple Mutation" uses a generic improvement instruction, while "Full" uses our structured, multi-part master prompt. Scores are AUC.

|  | *Detective* | *Zork1* | *Balances* |
|---|---|---|---|
| **EvoTest (Full Structured Evolution)** | **0.94** | **0.14** | **0.32** |
| EvoPrompt | 0.65 | 0.10 | 0.24 |
| EvoTest (w/ Simple Mutation) | 0.65 | 0.07 | 0.20 |
| EvoTest (w/o Prompt Evolution) | 0.52 | 0.05 | 0.16 |

Table 6: Comparison of AUC scores in a setting where the Actor LLM is fixed to `qwen/qwen3-32b` for all methods.

| Method | Actor LLM | Evolver LLM | Detective | Zork1 | Balances | Avg. |
|---|---|---|---|---|---|---|
| SFT (online) | qwen3-32b | N/A | 0.40 | 0.11 | 0.22 | 0.24 |
| GRPO (online) | qwen3-32b | N/A | 0.55 | 0.07 | 0.30 | 0.31 |
| EvoTest (Ours) | qwen3-32b | qwen3-32b | 0.68 | 0.10 | 0.28 | 0.35 |
| EvoTest (Ours) | qwen3-32b | openai/o3 | **0.78** | **0.12** | **0.31** | **0.40** |

run, our learning step is a single API call to an LLM, which takes about 20-30 seconds. We provide a more formal analysis of the computational complexity of EvoTest in Appendix E

**Impact of the LLM of Evolver Agent.** To assess the sensitivity of our framework to the Evolver's reasoning capabilities, we conducted an ablation study on its underlying LLM (Table 4). The results reveal a clear correlation between model scale and agent performance; more powerful models like `openai/o3` consistently yield higher scores, likely due to their superior ability to distill complex strategic insights from raw episode transcripts. Notably, even with a significantly smaller model such as `qwen3-8b`, performance remains substantially above the non-learning *Static* baseline. This finding demonstrates the robustness of the EvoTest framework: while a more capable Evolver LLM acts as a performance amplifier, the fundamental act-evolve loop is effective in its own right.

**Ablation on the Structure of Prompt Evolution.** To further isolate the contribution of our prompt-evolving logic, we create a baseline, "EvoTest (w/ Simple Mutation)," which replaces our detailed Evolver prompt with a generic instruction to simply "analyze the trajectory and generate an improved prompt." The results in Table 5 show that while this simpler mutation improves over no prompt evolution, it significantly underperforms our system, with the AUC score on Detective dropping from 0.94 to 0.65. This demonstrates that the performance gains are not just from evolving the prompt, but from the Evolver's sophisticated analysis, which is a core contribution of our framework.

**A Fairer Comparison with Fine-Tuning Methods.** In this setup, we normalize the underlying model capabilities by using `qwen/qwen3-32b` as the backbone for all compared methods. As shown in Table 6, we evaluate two versions of our method: one using `qwen/qwen3-32b` for both the Actor and Evolver, and a second version that pairs the `qwen/qwen3-32b` Actor with the stronger `openai/o3` Evolver. The results confirm that even when both components use `qwen/qwen3-32b`, EvoTest (Avg. AUC 0.35) outperforms the strongest weight-update baseline, GRPO (0.31). Furthermore, the performance leap when using a more capable Evolver (Avg. AUC 0.40) underscores the impact of the optimizer's reasoning ability within our framework.

## 6 CONCLUSION

We introduce the J-TTL benchmark to measure test time agent learning and proposed EvoTest, a novel evolutionary framework that improves agentic systems at test-time without gradients. By analyzing entire episode transcripts, EvoTest evolves the complete agent configuration—policy, memory, and hyperparameters—to rapidly adapt. Our experiments show EvoTest significantly outperforms strong baselines, including reflection, prompt optimization, and online fine-tuning. Its strength lies in using rich, narrative feedback for credit assignment, a far more data-efficient paradigm than relying on sparse rewards. This work provides a concrete step toward building truly autonomous agents that learn and self-improve from experience.

ACKNOWLEDGEMENTS

This research is supported by the Ministry of Education, Singapore, under the Academic Research Fund Tier 2 (FY2025) (Grant MOE-T2EP20124-0009)

ETHICS STATEMENT

All authors of this paper have read and adhered to the ICLR Code of Ethics. Our work focuses on foundational research into the learning capabilities of AI agents within simulated, text-based environments. We have identified and considered the following potential ethical dimensions of this research:

**Inherited Bias in Language Models** Our framework, along with the baselines, utilizes large language models (LLMs) as backbones. It is well-documented that LLMs can inherit and amplify societal biases present in their training data. While the fictional context of the Jericho games is unlikely to surface common social biases, we acknowledge that the agents' generated language and decisions are fundamentally shaped by the underlying models. Our research does not introduce new sources of bias but operates within the existing limitations of current LLM technology.

**Dual-Use and Long-Term Implications** Research into self-improving autonomous agents contributes to a long-term vision of more capable and independent AI. Such technology could, in the distant future, have dual-use potential. However, our work is situated at a very early, foundational stage and is confined to a controlled, non-physical, and non-real-world gaming environment. The primary goal is to understand and measure test-time learning in a sandboxed setting, which is a critical step for developing safer and more reliable AI systems.

**Research Integrity** This research does not involve human subjects, personal data, or any form of deception. The datasets (Jericho games) and software are publicly available. We have been transparent about the use of LLMs for language polishing in the manuscript, as detailed in Appendix B. We have no conflicts of interest that could have influenced the results or their interpretation.

REPRODUCIBILITY STATEMENT

We are committed to the full reproducibility of our work. To this end, we have made extensive efforts to document our methodology and provide all necessary artifacts. The primary resources for reproducing our results are detailed in the appendices.

Our complete source code, which includes the implementation of the EvoTest framework, the J-TTL benchmark setup, and all baseline methods described in the paper, is provided in `https://github.com/yf-he/EvoTest`). **Appendix C** provides comprehensive implementation details, including the specific Jericho games used, the environment configuration (e.g., episode step limits), hardware requirements, and the exact model identifiers for all LLMs used. **Appendix D)** offers detailed descriptions of all baseline methods, enabling a faithful re-implementation of our comparisons. The core of our method's learning mechanism, the Evolver Agent's master prompt, is provided in its entirety in **Appendix H)**. Further experimental details, including a controlled comparison on the `Qwen3-32B` backbone, are available in **Appendix L)**. All key hyperparameters and random seeds will be included in our public code release to ensure deterministic replication. We believe these resources provide a clear and complete pathway to reproduce our experiments and validate our findings.

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

## CONTENTS

## A   LIMITATIONS

Our work introduces a novel evolutionary framework for test-time learning, but it is important to acknowledge the inherent trade-offs and limitations of this approach.

**The Drawback of Not Modifying Neural Weights.** Our framework's foundation is its gradient-free nature, which is key to its practicality and data efficiency at test time. This strategic decision, however, introduces fundamental trade-offs by constraining learning to the symbolic level, rather than the parametric level of the model's weights. 1) Bounded by Pre-trained Capabilities: Our approach, like other LLM-based agents, assumes the backbone model possesses latent knowledge sufficient for the task. The central challenge we address is not the knowledge itself, but the development of an efficient test-time learning mechanism to surface and apply this knowledge on the fly. EvoTest operates as a high-level orchestrator; it learns to discover and refine the existing capabilities of its backbone LLM. However, it cannot instill fundamentally new, low-level reasoning patterns that lie outside the model's pre-trained knowledge base. For instance, if a game required a form of complex spatial reasoning that the LLM has never mastered, no amount of prompt evolution or memory management could create this capability from scratch. Fine-tuning, in contrast, holds the long-term promise of teaching a model truly novel skills. The performance of EvoTest is therefore capped by the inherent intelligence of its frozen backbone. Future work could explore hybrid approaches that use EvoTest for rapid, episode-to-episode adaptation while employing very slow, background fine-tuning to gradually enhance the agent's core competencies over hundreds of sessions. 2) Task-Specific Overfitting vs. Generalizable Skill Acquisition: The J-TTL benchmark and our method are designed for rapid improvement on a single task. The highly specific strategies and "verbal guardrails" that EvoTest learns (e.g., `unlock door with key`) are a form of strategic overfitting to the current task instance. This is highly effective for the TTL setting, but the learned knowledge is brittle and may not generalize to even minor variations in the task environment (e.g., if the key were in a different room). In contrast, traditional RL and fine-tuning, when applied across a diverse distribution of tasks, aim to learn more robust and generalizable policies. A key direction for future research is to investigate how the symbolic strategies evolved by EvoTest for specific tasks could be collected and abstracted to build a library of general-purpose skills.

**Dependency on a Powerful Evolver Agent.** Our framework decouples acting from adaptation, but in doing so, it creates a strong dependency on the reasoning capabilities of the Evolver LLM. As our ablation in Table 4 demonstrates, the quality of the evolutionary step is correlated with the power of the model performing the analysis. This means that the success of the entire system hinges on access to a capable-and potentially expensive-"optimizer" model.

**Complexity of the Evolutionary Search Space.** The agentic configuration $\chi = (p, M, h, u)$ forms a vast and complex combinatorial search space. Our current framework employs a simple $(1 + m)$ evolutionary strategy with UCB selection, which, while effective, is a relatively simple search heuristic. It may be prone to converging on local optima, especially in more complex games. Future work could explore more sophisticated population-based evolutionary algorithms, quality-diversity methods, or techniques from automated program synthesis to more effectively navigate this complex strategic landscape.

## B USE OF LARGE LANGUAGE MODELS FOR LANGUAGE POLISHING

In the preparation of this manuscript, we used Large Language Models (LLMs) as a writing assistance tool to enhance language, clarity, and readability. This usage was strictly confined to polishing text that was already drafted by the human authors.

Our process was interactive. After writing the core content, we used LLMs with specific prompts to refine the text. These prompts included requests to "check for grammatical errors," "rephrase this sentence for clarity," "make this paragraph more concise," or "suggest alternative phrasing to improve flow."

All suggestions generated by the LLM were critically reviewed, and the human authors retained full editorial control, making all final decisions regarding the manuscript's content and wording. The LLMs were not used to generate any scientific ideas, experimental results, data analysis, or other core intellectual contributions of the paper. The role of the LLM was analogous to that of an advanced grammar and style checker, and all research and conclusions presented are entirely the work of the authors.

## C IMPLEMENTATION DETAILS

### C.1 DATASETS AND ENVIRONMENT

- **Datasets:** Our evaluation is conducted on six publicly available Interactive Fiction (IF) games from the Jericho suite (Hausknecht et al., 2020). These games were chosen to represent a diverse range of puzzle structures and difficulties: `Detective`, `Library`, `Zork1`, `Zork3`, `Balances`, and `Temple`.
- **Environment:** All experiments are run using Jericho's standard Python API, `FrotzEnv`. For each game, a test-time learning session consists of $K = 50$ consecutive episodes. To ensure fair comparison, each episode is capped at a maximum of $T = 110$ interaction steps. The game environment is reset to its identical initial state after each episode, meaning any performance gains are solely attributable to the agent's learning algorithm. All textual observations from the game are converted to lowercase before being processed by the agent. All methods, including our own and all baselines, operate under these identical environmental constraints.

### C.2 HARDWARE

There is a significant difference in hardware requirements between the gradient-free and gradient-based methods evaluated.

- **Gradient-Free Methods:** EvoTest and all non-weight-update baselines (e.g., Static, Memory, RAG, Reflexion, TextGrad, EvoPrompt) do not require specialized local hardware. Their learning steps are executed via API calls to external LLMs. As such, these methods can be run on a standard machine with a CPU and a stable internet connection.
- **Gradient-Based Methods:** The weight-update methods, SFT (online) and GRPO (online), have substantial hardware demands. The online fine-tuning and policy gradient updates were performed on a dedicated cluster equipped with **4 NVIDIA H100 GPUs**. This hardware is necessary to accommodate the model's weights, gradients, and optimizer states in memory and to complete the training step in a reasonable time frame.

### C.3 REPRODUCIBILITY

We are committed to ensuring that our results are fully reproducible. To this end, we will make our code and experimental artifacts publicly available.

- **Code Release:** The complete source code for the EvoTest framework, the J-TTL benchmark setup, and all baseline implementations will be released at a public GitHub repository upon publication. The current code is available for review at https://github.com/yf-he/EvoTest.
- **Model and Environment Identifiers:** We have specified the exact model identifiers used for all roles and baselines (e.g., `google/gemini-2.5-flash`, `openai/o3-2025-04-16`). We

will also provide the version of the Jericho library and specific game files used in our experiments to prevent discrepancies arising from environment updates.

- **Configuration and Prompts:** The initial configuration files, including the generic starting prompts for the Actor Agent and the detailed master prompts used to guide the Evolver Agent, will be included in the code release. This is essential for reproducing the evolutionary trajectory of the agents.

- **Full Experimental Logs:** To facilitate detailed analysis and verification, we will release the complete logs from all our experimental runs. These logs will include the per-episode transcripts, the sequence of evolved configurations (prompts, hyperparameters, etc.), UCB selection scores, and final episode scores for every method on every game.

- **Seeds and Hyperparameters:** All random seeds used for LLM sampling and environment initialization will be provided. Additionally, all fixed hyperparameters, such as the UCB exploration constant $\beta$ and the number of children $m$ generated per evolution step, will be documented in the repository.

---

**Algorithm 1** EvoTest: Evolutionary Test-Time Learning

---

1: **Input:** Number of episodes $K$, initial configuration $\chi^{(1)}$
2: Initialize history of returns $\mathcal{H} \leftarrow \emptyset$
3: **for** $e = 1, \ldots, K$ **do**
4:                                                   ▷ –== Acting Phase ==–
5:     Select configuration $\chi^{(e)}$ for the current episode.
6:     **Actor Agent:** Execute episode using $\chi^{(e)}$ to get trajectory $\tau^{(e)}$ and return $R(e)$.
7:     Update statistics for $\chi^{(e)}$: $n(\chi^{(e)}) \leftarrow n(\chi^{(e)}) + 1$, update $\hat{\mu}(\chi^{(e)})$.
8:                                                ▷ –== Evolution Phase ==–
9:     **Evolver Agent:**
10:     Update Memory: $M^{(e+1)} \leftarrow \text{UpdateMemory}(M^{(e)}, \tau^{(e)})$.
11:     Generate child configurations $C^{(e+1)} = \{\tilde{\chi}_1, \ldots, \tilde{\chi}_m\}$ by applying evolutionary operators (prompt mutation, etc.) to $\chi^{(e)}$ and incorporating $M^{(e+1)}$.
12:                                                ▷ –== Selection Phase ==–
13:     Select next configuration $\chi^{(e+1)}$ from $\{\chi^{(e)}\} \cup C^{(e+1)}$ using the UCB rule (Eq. 5).
14: **end for**
15: **Return:** Sequence of episode returns $\{R(1), \ldots, R(K)\}$.

---

## D  BASELINES.

To rigorously evaluate the performance of EvoTest on the J-TTL benchmark, we compare it against a comprehensive suite of baseline methods. These baselines are organized into four distinct categories based on their underlying learning strategy: memory-based methods, reflection-based methods, automated prompt optimization methods, and weight-update methods. We also include a non-learning static agent to establish a zero-shot performance floor. For all comparisons, methods are allocated the same step budget per episode and, where applicable, utilize the same backbone LLMs to ensure a fair and controlled evaluation environment.

**Non-Learning Baseline.**

- **Static:** This agent serves as the fundamental, non-learning baseline. It operates with a single, fixed configuration—including a generic, hand-crafted prompt (e.g., "Explore the environment and try to score points.") and default hyperparameters—for the entire duration of the test session. It performs no updates between episodes and has no mechanism for cross-episode memory. Its purpose is to measure the zero-shot performance of the backbone LLM on the task and establish a reference point against which all learning-based improvements can be quantified. Any variation in its score across episodes is attributable solely to the inherent stochasticity of the LLM's generation process.

**Memory-based Methods.** These methods learn by accumulating and accessing information from past interactions. However, they do not modify the agent's core strategic prompt or its decision-making logic.

- **Memory (Full History):** This method attempts to learn by providing the agent with maximum historical context. After each episode, the complete transcript of that episode is appended to a growing history of all previous transcripts in the session. This full session history is then placed into the LLM's context window for the subsequent episode. The primary learning mechanism is in-context learning, where the LLM is expected to identify patterns and successful strategies from the raw text of past attempts. Its main limitation is the finite context window of the LLM; when the session history exceeds the context limit, the oldest parts of the history are automatically truncated.
- **RAG (Retrieval-Augmented Generation):** This agent enhances its decision-making by actively retrieving relevant information from past experiences. All trajectories from the current session are stored in a vector database. At each step within an episode, the agent's current observation is used to query this database to find the most similar or relevant snippets from past trajectories. These retrieved snippets, which may contain successful or failed state-action sequences from similar situations, are then injected into the prompt as additional context. This allows the agent to dynamically access pertinent past knowledge without being constrained by a fixed context window. The base prompt and hyperparameters, however, remain static throughout the session.

**Reflection-based Methods.** This category includes methods that use an LLM to generate high-level textual analyses of past performance, which are then used to guide future behavior.

- **Summary:** This agent learns by creating a condensed narrative of its experiences. After each episode, an LLM is prompted to progressively summarize the entire history of all past transcripts. This summary is updated after every episode to incorporate the latest attempt, creating a concise, high-level overview of the session's progress, including key discoveries and persistent challenges. This condensed summary is then prepended to the agent's prompt for the next episode, aiming to provide strategic context without consuming the entire context window with raw transcripts.
- **Reflexion** (Shinn et al., 2023): A prominent "verbal reinforcement learning" baseline. After each episode concludes, the agent reflects on its performance by analyzing the trajectory transcript. It generates a structured self-reflection that identifies specific failures, hypothesizes their root causes, and formulates an explicit, actionable plan to avoid those mistakes in the future (e.g., "I got stuck in the kitchen because I kept trying to 'open the locked pantry'. In the next attempt, I must first 'find the pantry key' in the living room."). This textual reflection is then added to the agent's prompt, accumulating over episodes to build a rich, strategy-focused memory that directly informs future decision-making.

**Automated Prompt Optimization Methods.** These methods focus on iteratively refining the agent's core policy by directly modifying its guiding system prompt.

- **TextGrad** (Yuksekgonul et al., 2024): This method is adapted for our test-time learning setting by treating the prompt as a set of "textual parameters" to be optimized. After each episode, the trajectory and the prompt that generated it are passed to a separate "optimizer" LLM. This optimizer generates a "textual gradient"—a short, critical analysis describing a flaw in the prompt and suggesting a direction for improvement. A subsequent LLM call then "applies" this gradient by editing the original prompt based on the critique. This creates a refined prompt for the next episode, directly evolving the agent's high-level strategy.
- **Promptbreeder** (Fernando et al., 2023) & **EvoPrompt** (Guo et al., 2024): These two methods are adapted from their original formulations to our sequential, single-session setting. Both employ an evolutionary algorithm to optimize a population of prompts. The process begins with a set of initial seed prompts. For each episode, a prompt is selected from the population, and its performance is evaluated based on the final episode score, which serves as its "fitness." After the episode, this fitness score is used to guide evolutionary operations. High-performing prompts are selected for "mutation" (where an LLM makes small modifications to the prompt) and "crossover" (where an LLM combines two successful prompts). The prompt for the next episode is then selected from

this newly evolved population. This creates a competitive, population-based search for the most effective guiding instruction.

**Weight-Update Methods (Online Fine-Tuning).** In contrast to the gradient-free methods above, this category includes baselines that directly modify the weights of the backbone LLM via online fine-tuning. These methods represent the traditional approach to model adaptation.

- **SFT (online):** This agent learns by imitating its own past behavior. After each episode, the trajectory is converted into a dataset of (state, action) pairs. The backbone LLM is then fine-tuned on this dataset using a standard Supervised Fine-Tuning (SFT) objective. This update adjusts the model's weights to increase the likelihood of generating the actions it took in the previous episode, given the same states. This approach reinforces the entire trajectory, which can be effective for successful runs but risks strengthening poor decision-making patterns from failed attempts.
- **GRPO (online):** (Shao et al., 2024) This agent uses a gradient-based Reinforcement Learning (RL) approach to update its policy. After each episode, the trajectory's state-action pairs and their associated rewards are used to compute a policy gradient. The model's weights are then updated to "reinforce" actions that led to positive rewards and suppress those that did not. This allows for more nuanced, reward-guided credit assignment than SFT. However, its effectiveness is highly dependent on the quality and density of the reward signal, and it is computationally intensive, requiring significant GPU resources for backpropagation through the model.

# E   COMPUTATIONAL COMPLEXITY ANALYSIS

In this section, we provide a more formal analysis of the computational complexity of EvoTest compared to the baselines, focusing on the cost per test-time learning cycle (one episode of acting and one phase of learning). We define the following variables for our analysis:

- $K$: The total number of episodes in a session.
- $T$: The maximum number of steps per episode.
- $C_t$: The context length (in tokens) provided to the Actor LLM at step $t$.
- $L_a$: The average length (in tokens) of a generated action.
- $L_o$: The average length (in tokens) of an environment observation.
- $\tau_L = T \cdot (L_o + L_a)$: The approximate total length of an episode transcript.
- $m$: The number of child configurations generated by the Evolver in EvoTest.
- $P_{\text{actor}}$: The number of parameters in the actor LLM.
- $P_{\text{evolver}}$: The number of parameters in the evolver/optimizer LLM.
- $d$: The hidden dimension of the actor LLM's transformer architecture.

We model the cost of an LLM forward pass for generating $L_{\text{out}}$ tokens from an input of $L_{\text{in}}$ tokens as $\text{Cost}_{\text{LLM}}(L_{\text{in}}, L_{\text{out}})$. This cost is primarily dependent on the model size and the total number of tokens processed.

**Complexity of EvoTest.** The cost of a single EvoTest cycle can be decomposed into the Acting Phase and the Evolution Phase.

**1. Acting Phase:** In each episode, the Actor Agent takes $T$ steps. At each step $t$, it queries the backbone LLM.

$$\text{Cost}_{\text{Act}} = \sum_{t=1}^{T} \text{Cost}_{\text{LLM}}(C_t, L_a) \approx T \cdot \text{Cost}_{\text{LLM}}(\bar{C}, L_a) \qquad (6)$$

where $\bar{C}$ is the average context length. This cost is dominated by $T$ forward passes through the actor model.

**2. Evolution Phase:** After the episode, the Evolver Agent performs a single, large query to generate new configurations. The input is the full episode transcript ($\tau_L$).

$$\text{Cost}_{\text{Evolve}} = \text{Cost}_{\text{LLM}}(\tau_L, L_{\text{config}}) \qquad (7)$$

where $L_{\text{config}}$ is the length of the generated configuration text. The UCB update step is $\mathcal{O}(m)$, which is negligible compared to the LLM call.

The total cost for one cycle of EvoTest is thus:

$$\text{Cost}_{\text{EvoTest}} = T \cdot \text{Cost}_{\text{LLM}}(\bar{C}, L_a) + \text{Cost}_{\text{LLM}}(\tau_L, L_{\text{config}}) \tag{8}$$

This cost is entirely composed of LLM forward passes, which can be served via APIs without requiring local GPU memory for gradients.

**Complexity of Baselines.**

- **Static/Memory:** The cost is simply the acting phase, $\text{Cost}_{\text{Act}}$. These are the most efficient but least effective methods.
- **Reflexion/EvoPrompt:** These methods have a similar complexity profile to EvoTest. Their learning phase also consists of a single large LLM call that takes the transcript $\tau_L$ as input to generate a reflection or a new prompt. Their total cost is structurally identical to Equation 3.
- **Online RL (GRPO):** This is where the complexity profile differs fundamentally. The cycle consists of an acting phase and a weight-update phase.

**1. Acting Phase (RL):** The cost is identical to other methods: $\text{Cost}_{\text{Act}} = T \cdot \text{Cost}_{\text{LLM}}(\bar{C}, L_a)$.

**2. Weight-Update Phase (RL):** This phase involves backpropagation to update the model weights. The computational cost of a training step for a transformer model is approximately proportional to the number of parameters and the total sequence length processed. For an entire episode trajectory of length $T$, this cost is:

$$\text{Cost}_{\text{Update}} \approx \mathcal{O}(P_{\text{actor}} \cdot T) \tag{9}$$

This cost reflects the computation for a full forward and backward pass through the trajectory to compute gradients. More critically, this step has substantial hardware requirements. The GPU VRAM must be large enough to store:

- Model Weights: $\mathcal{O}(P_{\text{actor}})$
- Gradients: $\mathcal{O}(P_{\text{actor}})$
- Optimizer States (e.g., Adam): $\mathcal{O}(2 \cdot P_{\text{actor}})$
- Activations: $\mathcal{O}(T \cdot d \cdot \text{batch\_size})$

The memory for activations scales with the episode length $T$, making online fine-tuning on long trajectories very demanding. The total cost for one cycle of online RL is:

$$\text{Cost}_{\text{RL}} = T \cdot \text{Cost}_{\text{LLM}}(\bar{C}, L_a) + \mathcal{O}(P_{\text{actor}} \cdot T) \tag{10}$$

**Comparative Summary.** As shown in Table 7, EvoTest's architecture trades the expensive, hardware-intensive backpropagation step of online RL for an additional LLM forward pass. While a large LLM call is not free, it is computationally cheaper than a full fine-tuning pass and, most importantly, can be offloaded to an API. This obviates the need for specialized local hardware (high-VRAM GPUs) and makes EvoTest a more data-efficient and practical solution for the test-time learning paradigm.

Table 7: Complexity comparison of a single learning cycle.

| Aspect | EvoTest (Ours) | Online RL (GRPO) |
|---|---|---|
| **Acting Cost** | $T \cdot \text{Cost}_{\text{LLM}}$ | $T \cdot \text{Cost}_{\text{LLM}}$ |
| **Learning Cost** | $\text{Cost}_{\text{LLM}}(\tau_L, L_{\text{config}})$ | $\mathcal{O}(P_{\text{actor}} \cdot T)$ |
| **Mechanism** | Gradient-Free (Forward Pass) | Gradient-Based (Backprop.) |
| **Hardware Req.** | CPU + Network | High-VRAM GPU |
| **Scalability Driver** | API Latency | GPU Compute & Memory |

## F    DETAILED MULTI-SEED EXPERIMENTAL RESULTS

This section provides the detailed multi-seed results mentioned in our response to the reviewers. The table 8 reports the mean Area Under the Curve (AUC) and standard deviation over 5 random seeds for all methods using the `google/gemini-2.5-flash` backbone, validating the robustness of our findings.

Table 8: Detailed comparison of Area Under the Curve (AUC) scores on the J-TTL benchmark, showing **mean ± std over 5 seeds** for the `google/gemini-2.5-flash` backbone. Higher values indicate better overall performance. The best performance in each column is highlighted in **bold**. These results robustly confirm that **EvoTest** consistently outperforms all baselines.

| Method | Detective | Library | Zork1 | Zork3 | Balances | Temple | Avg. |
|---|---|---|---|---|---|---|---|
| *Non-learning Baseline* | | | | | | | |
| Static | $0.22 \pm .02$ | $0.14 \pm .01$ | $0.03 \pm .01$ | $0.05 \pm .01$ | $0.11 \pm .02$ | $0.08 \pm .01$ | $0.11 \pm .01$ |
| *Memory-based & Reflection-based Methods* | | | | | | | |
| Memory | $0.56 \pm .02$ | $0.19 \pm .02$ | $0.04 \pm .01$ | $0.06 \pm .02$ | $0.14 \pm .02$ | $0.11 \pm .01$ | $0.13 \pm .01$ |
| RAG | $0.35 \pm .03$ | $0.25 \pm .03$ | $0.07 \pm .02$ | $0.10 \pm .02$ | $0.19 \pm .03$ | $0.16 \pm .02$ | $0.18 \pm .02$ |
| Summary | $0.46 \pm .04$ | $0.34 \pm .03$ | $0.11 \pm .02$ | $0.16 \pm .03$ | $0.26 \pm .03$ | $0.22 \pm .03$ | $0.26 \pm .03$ |
| Reflexion | $0.59 \pm .05$ | $0.42 \pm .04$ | $0.09 \pm .02$ | $0.26 \pm .03$ | $0.31 \pm .03$ | $0.30 \pm .04$ | $0.33 \pm .03$ |
| *Automated Prompt Optimization Methods* | | | | | | | |
| TextGrad | $0.62 \pm .05$ | $0.46 \pm .04$ | $0.12 \pm .03$ | $0.29 \pm .04$ | $0.17 \pm .03$ | $0.24 \pm .03$ | $0.32 \pm .03$ |
| Promptbreeder | $0.64 \pm .04$ | $0.48 \pm .04$ | $0.11 \pm .02$ | $0.30 \pm .04$ | $0.24 \pm .03$ | $0.31 \pm .03$ | $0.35 \pm .03$ |
| EvoPrompt | $0.64 \pm .04$ | $0.48 \pm .03$ | $0.13 \pm .02$ | $0.31 \pm .03$ | $0.25 \pm .03$ | $0.28 \pm .04$ | $0.35 \pm .03$ |
| *Weight-Update Methods (Online Fine-Tuning)* | | | | | | | |
| SFT (online) | $0.41 \pm .06$ | $0.31 \pm .05$ | $0.11 \pm .04$ | $0.19 \pm .04$ | $0.23 \pm .05$ | $0.20 \pm .04$ | $0.24 \pm .04$ |
| GRPO (online) | $0.58 \pm .05$ | $0.39 \pm .04$ | $0.08 \pm .03$ | $0.23 \pm .04$ | $0.31 \pm .04$ | $0.25 \pm .03$ | $0.31 \pm .03$ |
| **EvoTest (Ours)** | $\mathbf{0.93} \pm .02$ | $\mathbf{0.78} \pm .03$ | $\mathbf{0.15} \pm .02$ | $\mathbf{0.36} \pm .02$ | $\mathbf{0.33} \pm .02$ | $\mathbf{0.32} \pm .02$ | $\mathbf{0.48} \pm .01$ |

## G    THE TEST-TIME LEARNING PROBLEM: A DETAILED ANALYSIS

This section provides a more detailed analysis of the core research problem addressed by our J-TTL benchmark, illustrating why traditional learning paradigms fail and motivating the design of EvoTest.

### G.1    THE TASK: LEARNING AND MASTERY IN TEXT-ADVENTURE GAMES

The task given to the agent in our J-TTL benchmark is designed to be a challenging test of on-the-fly learning and adaptation.

**The Environment.** The agent interacts with a classic text-based adventure game from the Jericho suite (e.g., *Detective*, *Zork1*). In these games, the entire world is described through text. The agent receives textual observations (e.g., `"You are in the Chief's office.  A piece of white paper is on the desk."`) and must issue text commands (e.g., `TAKE PAPER`) to act. These games are notoriously difficult for AI due to complex puzzles, long-range planning dependencies, sparse rewards, and irreversible consequences.

**The Goal.** The agent's goal is structured at two levels:

- **The Episodic Goal (Maximize Score):** Within a single playthrough (one "episode"), the agent's objective is to take actions that maximize its final score, which is awarded for discovering areas, solving puzzles, and advancing the plot.
- **The Learning Goal (Improve Across Episodes):** The ultimate task is to play the *same game repeatedly* and demonstrate learning by progressively increasing its final score from one episode to the next. The agent must use its experience from failed or suboptimal attempts to build a better strategy for subsequent attempts.

## G.2 A Concrete Failure Case: The Navigation Loop

To illustrate why this task is hard, consider an agent attempting to play *Detective*. It might correctly execute `GO WEST`, but from the next location, it gets stuck by repeatedly attempting `GO WEST` again, an invalid move the game rejects with `"You can't go that way."` This simple failure highlights the limitations of existing methods.

**The Static Agent.**  A non-learning agent with a fixed, generic prompt has no mechanism to correct this error. It will likely repeat the same mistake in every episode, resulting in a flat, low-scoring performance curve. It cannot adapt.

**The Online SFT Agent.**  A more sophisticated agent might use Supervised Fine-Tuning (SFT) on its prior trajectory. Our SFT baseline intelligently filters for "positive" actions (those that yielded a score increase). This approach fails for two reasons:

- In the low-scoring episode where the agent got stuck, it generated very few, if any, positive actions. The dataset for fine-tuning is therefore either empty or extremely small. With no good data to learn from, the agent cannot improve.
- Many critical actions in these games are "neutral" and provide no immediate reward (e.g., `UNLOCK DOOR WITH KEY`). An SFT agent that only trains on score-increasing actions will never learn these essential intermediate steps, rendering it incapable of solving complex puzzles.

**The Online RL Agent.**  A Reinforcement Learning (RL) agent receives a reward of '0' for the invalid `GO WEST` action. In a sparse-reward environment like Jericho, this signal is incredibly weak and ambiguous. It is indistinguishable from the 'reward=0' received for a neutral but necessary action. A single gradient update based on this noisy signal is insufficient to meaningfully correct the agent's policy for the next attempt, demonstrating a failure of credit assignment in a low-data regime.

## G.3 The EvoTest Solution

EvoTest is designed to overcome these failures. Its **Evolver Agent** analyzes the **entire episode transcript**, not just scalar rewards or positive actions.

**It Learns from Failure:** Unlike SFT, EvoTest learns most effectively from failures. The Evolver semantically identifies the unproductive loop by reading the game's textual feedback (`"You can't go that way."`) paired with the repeated action and recognizes it as a problem to be solved.

**It Performs Whole-System Evolution:** Based on its analysis, the Evolver directly rewrites the agent's prompt, generating a targeted, structural edit to correct the error (e.g., adding a new rule: `Step 5: From the street, GO EAST to enter the Mayor's house.`). This is a far more direct and data-efficient learning mechanism than a small, gradient-based weight update.

## H The Evolver Agent's Master Prompt

The core of EvoTest's learning capability resides in the Evolver Agent, which is guided by a comprehensive "master prompt." This prompt structures the analysis of a completed episode transcript, enabling the Evolver's LLM to perform holistic, multi-faceted updates across the entire agentic configuration $\chi = (p, M, h, u)$. Unlike simpler approaches that only modify the policy prompt, our master prompt instructs the Evolver to act as a full-system optimizer, proposing changes to the agent's high-level strategy, its structured memory, its low-level decision-making parameters, and its internal tool-use logic.

The prompt is divided into four distinct parts, each targeting a specific component of the agentic system.

### EvoTest Master Prompt: Preamble and Context

```
You are an AI agent system optimizer. Your task is to analyze the transcript of a
text-adventure game session and generate a new, improved configuration for the next
agent that will play the same game. The goal is to help the agent score higher in
the next episode.

The agent's configuration has four components:
1.  A guiding prompt (the agent's high-level strategy).
2.  Memory updates (structured data for a success/failure database).
3.  Hyperparameters (like temperature, for decision-making).
4.  Tool-use routines (Python code for state abstraction and rules for memory access).

You will receive the previous guiding prompt and the full game history. Generate a
new, complete configuration by following the four parts below.

The LLM agent used the following guiding prompt (which may not be accurate; rewrite
it  if needed):
"{cur_prompt}"

Here is the history of that game session:
--- GAME HISTORY START ---
{cur_history_str}
--- GAME HISTORY END ---
{negative_section}
```

---

**EvoTest Master Prompt: Detailed Generation Instructions**

```
PART 1: Generate a new improved guiding prompt.
This prompt is the agent's high-level policy. Structure it clearly.
1.  Create a ``Walkthrough" or ``Essential Actions" section. Identify all useful
    actions from the history that led to score increases or were strictly necessary
    for game progression. Synthesize these into a clear, step-by-step plan. Be
    precise with action phrasing (e.g., ``unlock door with key" instead of ``use key").
2.  Create an ``Actions to Avoid" section. Identify actions that led to getting
    stuck, caused loops, produced repeated errors, or were clearly unproductive.
    List these as negative constraints or ``guardrails."
3.  If the agent has not yet won, create a final ``Exploration Plan" section.
    Brainstorm possible next steps. List rooms or objects that have been seen but
    not fully interacted with. Suggest a systematic approach for the agent to follow
    once the known walkthrough is complete (e.g., ``visit every room, systematically
    use LOOK, EXAMINE, SEARCH, and try actions like PUSH, PULL, READ on all objects.").

PART 2: Generate memory updates for the database.
Analyze the transcript and extract all state-action pairs that resulted in a score
increase. Format this as a JSON list of objects. This data will be added to the
agent's success memory.
- The `state_text` should be the full observation text right before the action was taken.
- The `action` is the command the agent gave.
- The `score_delta` is the positive score change.
If no new score increases were found, return an empty list `[]`.
Example:
[
  {{"state_text": ``<< closet >>\nyou are in a closet. ...", ``action": ``take pistol", ``score_delta": 10}},
  {{"state_text": ``<< living room >>\nyou are standing ...", ``action": ``get wood", ``score_delta": 10}}
]

PART 3: Suggest hyperparameter adjustments.
Analyze the agent's overall behavior to tune the exploration-exploitation balance.
Output a JSON object.
- If the agent was stuck in repetitive loops or seemed overly cautious, suggest
  increasing the temperature to encourage more diverse actions (e.g., `"temperature": 0.8`).
- If the agent's actions seemed chaotic, random, or deviated from a good existing
  plan, suggest decreasing the temperature to promote adherence to the strategy
  (e.g., `"temperature": 0.2`).
- If the agent's behavior was optimal and it followed the plan well, suggest no
  changes by returning an empty JSON object `{}`.
Example: `{{ ``temperature": 0.75 }}`

PART 4: Generate new tool-use routines.
4a. State Extractor Code: Generate a Python code block for a function named
`extract_state`. This function takes the full `game_history` string as input and returns
a short summary of key milestones completed.
- Base the logic on the presence of specific *strings from the game's output*, not
  the agent's actions.
- Keep the code simple to avoid bugs. Use `if/elif/else` with `in` checks.
- Do not include comments inside the function body.

4b. Memory Interaction Logic: Propose a single sentence to be added to the main
guiding prompt that refines how the agent should use its memory. This logic should
evolve to become more disciplined over time.
- Early-stage suggestion: ``Hint: You can check your memory of past successes for ideas
  in familiar situations."
- Mid-stage suggestion: ``Strategy: In any room you've visited before, consult your
  success memory for proven actions."
- Late-stage rule: ``Rule: Before every action, you MUST query your success memory
  for the current state and follow its suggestion if one exists."

Format your response as follows with NO additional text. The function name MUST be
`extract_state`.

[Your generated prompt here]
[Your JSON for memory updates here]
[Your JSON for hyperparameter adjustments here]

def extract_state(game_history):
    # [Return a string summarizing the current state]

[Your one-sentence memory interaction logic here]
```

---

## I CASE STUDY: LEARNING TO NAVIGATE THE LIBRARY

Analysis of EVOTEST's behavior in the *library* game reveals it is performing a form of Verbal Reinforcement Learning. In this paradigm, the agent's natural language prompt is its policy, the full episode transcript is the rich reward signal, and policy updates are semantic edits to the prompt. The evolution of this policy is detailed in the prompts shown in Figures 4 through 8.

**Episode 0: The Credit Assignment Problem.** The initial agent begins with a simple, high-level directive (Figure 4) and acquires the target biography, but gets stuck at the security alarm. It mis-

attributes the failure, falling into loops of re-locking the rare books room, demonstrating a classic credit assignment problem where a delayed, sparse negative signal is linked to the wrong proximate cause.

**Episode 1-3: Positive Verbal Policy Update.** The Evolver Agent analyzes the first transcript, identifies the successful action sequence for retrieving the biography, and distills it into a new heuristic. This **positive policy update** is evident in the updated prompt for Episode 1 (Figure 5) and its further refinement by Episode 3 (Figure 6). The Evolver performed credit assignment on the score-increasing events and codified the successful trajectory into the agent's strategy, effectively abstracting a reusable skill from a single experience.

**Episode 11: Latent Value Discovery.** By now, the evolved prompt has discovered a non-obvious but critical action: ASK TECHNICIAN ABOUT GATES (Figure 7). This command provides no immediate score increase but is a prerequisite for solving a later puzzle. This demonstrates **latent value discovery**. A traditional RL agent would struggle to find such an unrewarded action. EVOTEST uses semantic reasoning on the narrative—"an NPC is blocking an object; making them leave could be useful"—to identify a state with high future value, bypassing the need for numerical value propagation.

**Episode 49: Negative Policy Update and Guardrails.** The agent's policy is now a multi-stage plan. Crucially, it includes an "Actions to Avoid" section (Figure 8), warning against interacting with the security gates because it "causes an endless clarification loop." This is a **negative policy update**. The system has learned not just what to do, but what not to do, by identifying a frustrating failure pattern (a parser bug) from the transcript. It created a "verbal guardrail" to prevent this specific failure, a form of learning that is difficult to achieve with scalar rewards alone.

> **Prompt of Episode 0**
>
> ```
> Explore systematically and examine objects to make progress.
> ```

Figure 4: Case Study: Prompt of Episode 0.

# J   THE MEMORY COMPONENT IN PRACTICE: CONCRETE EXAMPLES

To illustrate precisely how the Evolver Agent constructs and utilizes memory, this section details the process using interactions from the *Detective* game. The memory is not a monolithic block of text; it is a structured database, programmatically populated by the Evolver after each episode.

## J.1   SUCCESS MEMORY: BUILDING A DATABASE OF WHAT WORKS

The Success Memory functions as a high-reward state-action lookup table. The Evolver parses the episode transcript for any step where the score increases and logs the preceding state and action.

For example, after a single episode, the Evolver identifies several score-increasing events from the trajectory log.

**Prompt of Episode 1**

```
Retrieve Graham Nelson biography efficiently:
Step-by-step actions that proved essential so far
1.  In lobby:  talk to reference librarian first { west to
Ground-Floor Stacks, ASK LIBRARIAN ABOUT GRAHAM NELSON and then
ASK LIBRARIAN ABOUT RARE.
2.  Return to lobby and GIVE ID CARD TO ATTENDANT; you receive
the Rare-Books KEY.
3.  Reach second-floor stacks (north, up, south) and UNLOCK RARE
BOOKS DOOR WITH KEY, OPEN DOOR, ENTER.
4.  TAKE BIOGRAPHY OF GRAHAM NELSON.
5.  Leave the room, CLOSE RARE BOOKS DOOR, then LOCK RARE BOOKS
DOOR WITH KEY.
6.  Back in lobby GIVE KEY TO ATTENDANT; he returns your ID
card.
Avoid
• Wandering in the dusty card-catalogue corner { it contains no
useful data.
• Trying to leave the library while still holding the key or
while the rare-books door is open/unlocked:  this triggers
alarms and extra trips.
• Vague commands such as \talk to attendant"; instead use verbs
ASK or GIVE.
Next places and ideas (not yet proven)
{ Northwest Duplicating Services room:  may let you photocopy
pages, reducing risk at security gate.
{ \Computer Room" east of 2F stacks.
{ The \Private" door north of lobby.
Systematically examine new rooms, objects, signage and NPCs;
look for legitimate ways to check out or copy the biography so
the alarm will not sound when passing the security gates.
```

Figure 5: Case Study: Prompt of Episode 1.

Prompt of Episode 3

```
Goal:  leave the library with the Biography of Graham Nelson
legitimately checked-out.
Essential steps that advanced the story and/or scored points
(use exactly these verbs { the game is picky):
1.  From the Lobby • ask attendant about book → he sends you to

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

(or \check out biography" / \borrow biography") • east → leave
the building without the alarm
Actions that wasted time or caused trouble { avoid these:
• Re-examining the dusty card-catalog drawers (they contain
unreadable parchment).
• \talk attendant" { the correct form is \ask attendant about
...".
• Dropping the biography in the stacks:  you must keep it and
check it out.
• Trying to exit before key is returned, door locked, and book
checked-out { triggers alarms.
Where to explore next if something is still missing:
Rooms not yet inspected in detail:  • Duplicating Services (NW
from Lobby)
• Private Room (N from Lobby)
• Computer Room (E from 2nd-floor stacks)
Systematically:  in every room LOOK, EXAMINE all obvious
objects, ASK any NPCs about topics that appear in room
descriptions, and INV and SCORE regularly to monitor progress.
```

Figure 6: Case Study: Prompt of Episode 3.

Prompt of Episode 11

ESSENTIAL WALKTHROUGH (all steps verified by score increases)
1. Gain entry to Rare-Books • ASK ATTENDANT ABOUT GRAHAM •
WEST to Ground-Floor Stacks • ASK LIBRARIAN ABOUT GRAHAM • ASK
LIBRARIAN ABOUT KEY • EAST to Lobby • ASK ATTENDANT ABOUT KEY •
GIVE ID CARD TO ATTENDANT→ receive Rare-Books key(+5)
2. Retrieve the Graham Nelson biography • WEST { NORTH { UP {
SOUTH to 2F Stacks • UNLOCK DOOR WITH KEY; OPEN DOOR; ENTER •
TAKE BIOGRAPHY(+5) • Return to 2F Stacks, CLOSE DOOR, LOCK DOOR
WITH KEY • NORTH { DOWN { SOUTH { EAST to Lobby • GIVE KEY TO
ATTENDANT(ID card returned)
3. Empty the Computer Lab and grab the encyclopedia • WEST {
NORTH { UP { SOUTH { EAST to Computer Lab • ASK TECHNICIAN ABOUT
GATES→ technician leaves(+2) • EXAMINE PRINTERS→ automatically
take Encyclopedia Frobozzica(+5)
4. Please the Reference Librarian • WEST { NORTH { DOWN {
SOUTH { EAST to Ground-Floor Stacks • GIVE ENCYCLOPEDIA TO
LIBRARIAN(+2)
Confirmed score so far: 19 points
ACTIONS TO AVOID • Getting stuck in the \which do you mean, the
howto manual or the security gates?" loop { when it appears,
simply type HOWTO MANUAL once and continue. • Trying to open
/ push / pry the \Private" north door. • Fussing with the
burned-out copier after the explosion. • Repeating LOOK/EXAMINE
on already-searched objects.
PLACES & IDEAS TO INVESTIGATE NEXT (unverified) • Read the HOWTO
MANUAL offered during the gate disambiguation; then SEARCH or
LOOK IN the opened security gate Tom is fixing. • Examine
paintings in both stairwells and the damaged picture on the
ground-floor stairs (MOVE, PUSH, PULL). • Re-visit Duplicating
Services after the smoke clears for anything the blast exposed.
• Scrutinise every object named in room texts (front doors,
notices, posters, card-catalog drawers, etc.). • ASK each NPC
ABOUT any new topic you uncover.
Systematic approach: in every room LOOK, then EXAMINE and
SEARCH every noun that appears. Try TAKE, OPEN, READ, PUSH,
PULL, TURN and ASK <npc> ABOUT <noun/topic> to expose further
clues.

Figure 7: Case Study: Prompt of Episode 11.

**Prompt of Episode 49**

```
ESSENTIAL ACTION LIST { VERIFIED TO 24 POINTS
1.  Borrow the Rare-Books key (+5) • In the Lobby:  GIVE ID CARD
TO ATTENDANT
2.  Confirm the book's location • WEST to Ground-Floor Stacks •
ASK LIBRARIAN ABOUT GRAHAM
3.  Retrieve the Graham Nelson biography (+5) • WEST { NORTH
{ UP { SOUTH to 2F Stacks • UNLOCK DOOR WITH KEY ; OPEN DOOR ;
ENTER • TAKE BIOGRAPHY • NORTH (leave) ; CLOSE DOOR ; LOCK DOOR
WITH KEY
4.  Return the key • NORTH { DOWN { SOUTH { EAST to Lobby • GIVE
KEY TO ATTENDANT (get ID card back)
5.  Clear the Computer Lab (+2 + 5) • WEST { NORTH { UP { SOUTH
{ EAST to Computer Lab • ASK TECHNICIAN ABOUT GATES (+2) •
EXAMINE PRINTERS (auto-take Encyclopedia Frobozzica, +5)
6.  Donate the Encyclopedia (+2) • WEST { DOWN { SOUTH { EAST {
WEST to Ground-Floor Stacks • GIVE ENCYCLOPEDIA TO LIBRARIAN
7.  Extra find in 2F Stacks (+5) • NORTH { UP { SOUTH to 2F
Stacks • SEARCH SHELVES (get \Debt of Honor")
Confirmed score so far:  24 points.
Avoid:  • Interacting with the security gates (causes an
endless clarification loop).  • Repeatedly fiddling with the
paintings/canvas until new clues appear.  • Non-standard verbs
(USE, TYPE, TALK TO, etc.).  • Trying to leave the library while
holding required books.
Unexplored areas worth checking once the above is complete (no
points verified yet):  • \Private" door in Lobby • Duplicating
Services room after explosion • Stairwell paintings (\grue"
canvas, damaged logo) • Card-catalog drawers and miscellaneous
shelving
After finishing the essentials, visit every room, LOOK, EXAMINE,
SEARCH, and ASK NPCs about any new noun you encounter, noting
any score changes.
Good luck!
```

Figure 8: Case Study: Prompt of Episode 49.

---

**Evolver Agent: Parsing Successes from Episode Log**

The Evolver's parser identifies the following successful interactions:

```
1. STATE: ``<< Chief's office >> ... You can see a piece of
white paper..."
   ACTION: ``read paper"
   REWARD: +10 points

2. STATE: ``<< closet >> ... there is a gun on the floor..."
   ACTION: ``get pistol"
   REWARD: +10 points

3. STATE: ``<< living room >> ... you see a battered piece of
wood..."
   ACTION: ``get wood"
   REWARD: +10 points
```

---

Based on these observations, the Evolver programmatically updates the `success_memory.json` file. This file stores a mapping from a hash of the state's descriptive text to the action that proved successful. The resulting database entries would look like this:

---

**Success Memory Database (`success_memory.json`)**

| State Hash (Key) | Stored Action (Value) | Score Delta |
|---|---|---|
| `hash("<< Chief's office...")` | `"read paper"` | +10 |
| `hash("<< closet >>...")` | `"get pistol"` | +10 |
| `hash("<< living room >>...")` | `"get wood"` | +10 |
| ... | ... | ... |

During the next episode, when the Actor Agent encounters the state `<< closet >>`, its memory-access routine will hash the state description, find a match in the database, and retrieve the action `"get pistol"`. This information is then used to augment the prompt, for example: **"Hint: In this exact situation before, the action 'get pistol' worked well."**

---

### J.2  FAILURE MEMORY: IDENTIFYING AND PRUNING UNPRODUCTIVE ACTIONS

The Failure Memory's goal is to prevent the agent from repeating obvious mistakes, especially getting stuck in loops. The Evolver identifies these patterns by detecting sequences of actions that result in no change to the game state or score.

From the provided logs, the Evolver can identify a wasted move:

---

**Evolver Agent: Parsing a Non-Productive Action**

The Evolver's analysis routine detects the following loop:

- **At Step t:** Agent is in state `<< hallway >> you are still in the hallway....`
- **Agent's Action:** `west`.
- **At Step t+1:** The environment returns the *exact same state description*: `<< hallway >> you are still in the hallway....`
- **Analysis:** The state did not change, and the score did not change. This action is flagged as unproductive for this state.

---

Unlike the Success Memory, this insight is not stored in a database to be queried. Instead, the Evolver uses it to directly mutate the agent's core policy—the prompt—by adding an explicit "verbal guardrail." This creates a more permanent and proactive change to the agent's strategy.

---

**Policy Prompt Mutation: Creating a Verbal Guardrail**

**Original Prompt Section (Episode N):**

```
## Strategy
- Explore systematically.
- Examine all objects.
```

**Evolved Prompt Section (Episode N+1):**

```
## Strategy
- Explore systematically.
- Examine all objects.

## Known Dead Ends / Wasted Actions
- In ``<< hallway >>", avoid the action ``west". It leads back to
the same room.
```

---

This example shows how EvoTest moves beyond simple trial-and-error. It performs semantic credit assignment on the narrative of the game, identifies a specific unproductive behavior, and encodes a rule to prevent it in the future. This form of learning, which prunes the search space by identifying and forbidding useless actions, is a key reason for its data efficiency compared to methods that rely solely on scalar rewards.

## K  EVOLUTION OF AGENT BEHAVIOR: KEY TRANSCRIPT SNAPSHOTS FROM *Detective*

To demonstrate how the agent's behavior evolves over a session, this section presents key moments from the transcripts of early, mid-stage, and late-stage episodes for the *Detective* game. Each snapshot includes the agent's guiding prompt at that stage, highlighting the direct link between the evolved policy and the agent's actions.

### K.1  EPISODE 3: EARLY EXPLORATION AND BASIC MISTAKES

In early episodes, the agent operates with a generic prompt and relies on broad exploration. Its behavior is characterized by discovering simple, high-reward actions while also making fundamental errors, such as getting stuck in loops.

---

**Guiding Prompt: Episode 3**

You are a detective playing a text-based game. Your goal is to solve the murder of the Mayor and maximize your score. Explore your surroundings, examine items, and talk to people to gather clues. Be systematic.

---

### K.1.1  KEY TRANSCRIPT STEPS: EPISODE 3

The agent successfully discovers initial points but then gets stuck in a simple navigational loop, revealing a lack of long-term strategy.

---

**Transcript Snapshot: Steps 2-9**

```
[STEP 2]
[OBS] confidential: detective was created by matt barringer...
[ACTION] west

[STEP 3]
[OBS] << closet >> ... there is a gun on the floor. better get it.
[ACTION] take pistol
[REWARD] +10

[STEP 4]
[OBS] taken. [your score has just gone up by ten points.]
[ACTION] east

[STEP 5]
[OBS] << chief's office >> ... go north or west.
[ACTION] north

[STEP 6]
[OBS] << outside >> ... to the east is a dead end. to the west
is ... street.
[ACTION] west
[REWARD] +10

[STEP 7]
[OBS] << outside >> ... to the north is a restraunt ... to the
east is ... home.
[ACTION] west

[STEP 8]
[OBS] you can't go that way. << outside >> ... north is a
restraunt...
[ACTION] west

[STEP 9]
[OBS] you can't go that way. << outside >> ... north is a
restraunt...
[ACTION] look
```

---

**Analysis:** The agent successfully finds points by taking the pistol (Step 3) and exploring west (Step 6). However, at Step 7, it fails to process the new information and tries to go 'west' again, an invalid move. It repeats this mistake at Step 8, demonstrating a classic failure mode of early-stage agents: getting stuck in a simple loop. This transcript provides a clear signal for the Evolver to create a rule about how to navigate this specific "outside" area.

### K.2 EPISODE 22: MID-STAGE EXECUTION OF A LEARNED PLAN

By the middle of the session, the prompt has evolved into a partial walkthrough. The agent can now flawlessly execute complex sequences it previously struggled with but may fail on details not yet encoded in its strategy.

---

**Guiding Prompt: Episode 22**

**Goal: Get to the Holiday Inn. Known Path:** 1. Office: 'read paper', 'west', 'take pistol', 'east', 'north'. 2. Mayor's House: Go east to living room for wood, then west to dining room for note. Then go north to exit. 3. Video/Music Stores: Navigate past them to reach the alley. **Avoid:** Going west multiple times at the street hub.

---

### K.2.1 KEY TRANSCRIPT STEPS: EPISODE 22

The agent expertly navigates the Mayor's house but fails at a crucial, un-memorized interaction due to incorrect action syntax.

```
Transcript Snapshot: Steps 8-13 and 30-32

[STEP 8]
[OBS] << mayor's house >> ... to the east is the living room ..
[ACTION] east

[STEP 9]
[OBS] << living room >> ... you see a battered piece of wood.
[ACTION] take wooden wood
[REWARD] +10

[STEP 10]
[OBS] taken. [your score has just gone up by ten points.]
[ACTION] west

[STEP 11]
[OBS] << mayor's house >> ... to the west is the dining room.
[ACTION] west

[STEP 12]
[OBS] << dining room >> ... see a note on the table.
[ACTION] take paper note
[REWARD] +10

[STEP 13]
[OBS] taken. [your score has just gone up by ten points.]
[ACTION] east
...
[STEP 30]
[OBS] << back of music store >> ... to the west there is a
dazed looking man ...
[ACTION] west

[STEP 31]
[OBS] << music store >> ... you need to ... kill him. best
chance: use your gun.
[ACTION] use gun

[STEP 32]
[OBS] what do you want to use the gun on?
[ACTION] dazed man
```

**Analysis:** The sequence from steps 8-13 shows perfect execution of a learned sub-plan, retrieving both items in the Mayor's house without error. This demonstrates successful memory and planning. The failure occurs later at Step 31. The prompt did not specify the exact syntax for using the gun, so the agent defaults to a generic 'use gun' command, which leads to a clarification question from the game and wastes a turn. The Evolver will parse this failure and refine the prompt to use the correct syntax: 'shoot dazed man with gun'.

### K.3 EPISODE 49: NEAR-PERFECT EXECUTION AND REFINEMENT

In late-stage episodes, the agent possesses a nearly complete walkthrough. Its behavior is precise, confident, and focused on executing the optimal path.

> **Guiding Prompt: Episode 49 (abridged)**
>
> Follow this exact sequence to score 330 / 360 and win... ... 8. Holiday Inn – capture • ...
> • north → elevator to 15th floor (+10) • west, north, west, north, north → Room 30 • enter
> room → automatic gun-fight, arrest killer (+100) • up → "info" room → *** YOU HAVE
> WON ***

### K.3.1 KEY TRANSCRIPT STEPS: EPISODE 49

The agent confidently executes the final, complex maze-like sequence of the game without a single
misstep.

> **Transcript Snapshot: Steps 43-49**
>
> ```
> [STEP 43]
> [OBS] << holiday inn >> ... you have to find it. to get
> started, go north.
> [ACTION] north
> [REWARD] +10
>
> [STEP 44]
> [OBS] << holiday inn 15th floor >> ... you can go east or west.
> [ACTION] west
> [REWARD] +10
>
> [STEP 45]
> [OBS] << hallway >> ... you can only go north.
> [ACTION] north
> [REWARD] +10
>
> [STEP 46]
> [OBS] << hallway >> ... you can go west or east.
> [ACTION] west
> [REWARD] +10
>
> [STEP 47]
> [OBS] << hallway >> ... you are getting close! you can only go
> north.
> [ACTION] north
> [REWARD] +10
>
> [STEP 48]
> [OBS] << hallway >> ... to the north is more hall.
> [ACTION] north
> [REWARD] +100
>
> [STEP 49]
> [OBS] << room # 30 >> ... after a harrowing gun battle you conk
> him on the head...
> to leave this darned game, go up.
> [ACTION] up
> ```

**Analysis:** This transcript shows the agent in its most effective state. The sequence 'west, north,
west, north, north' (Steps 44-48) is executed perfectly, demonstrating that the agent is no longer
exploring but is following a precise, validated plan. Each action leads directly to progress. This
flawless execution of a complex, non-obvious path is a hallmark of the converged EvoTest agent and
highlights the power of evolving a detailed, procedural policy.

Table 9: Comparison of AUC scores on a subset of games where all methods use `qwen/qwen3-32b` as the actor backbone LLM. EvoTest continues to outperform both weight-update baselines, highlighting the robustness of the evolutionary learning algorithm.

| Method | Detective | Zork1 | Balances | Avg. |
|---|---|---|---|---|
| SFT (online) | 0.40 | 0.11 | 0.22 | 0.24 |
| GRPO (online) | 0.55 | 0.07 | 0.30 | 0.31 |
| **EvoTest (Ours)** | **0.78** | **0.12** | **0.31** | **0.40** |

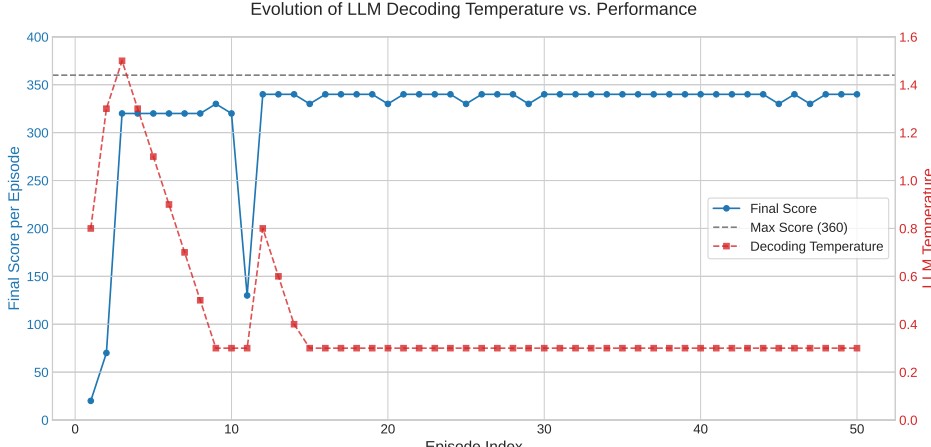

Figure 9: Evolution of the Actor's LLM temperature versus its performance on the *Detective* game.

## L  EXPERIMENT WITH QWEN3-32B BACKBONE

To provide a more direct and controlled comparison against the weight-update baselines, we conducted an additional experiment using `qwen/qwen3-32b` as the backbone LLM for all compared methods. In our main experiments, the weight-update methods (SFT and GRPO) use `qwen/qwen3-32b`, while EvoTest uses proprietary models. This supplementary experiment aims to normalize the underlying model capabilities to better isolate the effectiveness of the learning algorithms themselves.

In this setup, the Actor Agent in the EvoTest framework was switched from `google/gemini-2.5-flash` to `qwen/qwen3-32b`. The Evolver Agent continued to use `openai/o3-2025-04-16` to maintain its strong analytical capabilities, ensuring that EvoTest's performance reflects its architectural strengths rather than being limited by a weaker optimizer. The results, presented in Table 9, compare EvoTest against SFT and GRPO on a representative subset of games, with all three methods now relying on the same `qwen/qwen3-32b` backbone for their acting component.

The results confirm that EvoTest's performance advantage holds even in this controlled setting. As shown in Table 9, EvoTest surpasses both SFT (online) and GRPO (online) on all three tested games. The average AUC score for EvoTest (0.40) is substantially higher than GRPO (0.31) and SFT (0.24). This indicates that the superiority of our evolutionary approach is not merely an artifact of using a more powerful backbone model in the main experiments. Instead, it underscores the fundamental data efficiency of EvoTest's learning mechanism. By leveraging rich, narrative feedback from the entire episode transcript for whole-system evolution, EvoTest can make more significant and targeted improvements from a single experience than gradient-based methods relying on sparse scalar rewards.

Table 10: Comparison of AUC scores on a subset of games different LLM backbones. While stronger models moderately improve the Static agent's performance, the benefit is significantly greater for EvoTest, demonstrating the scalability of our learning framework.

| Method | Backbone LLM | Detective | Zork1 | Balances | Avg. |
|---|---|---|---|---|---|
| Static | `gemini-2.5-flash` | 0.21 | 0.03 | 0.11 | 0.12 |
| EvoTest | `gemini-2.5-flash` | 0.94 | 0.14 | 0.32 | **0.47** |
| Static | `claude-4-sonnet` | 0.23 | 0.04 | 0.12 | 0.13 |
| EvoTest | `claude-4-sonnet` | 0.95 | 0.16 | 0.35 | **0.49** |
| Static | `gemini-3-pro-preview` | 0.25 | 0.05 | 0.14 | 0.15 |
| EvoTest | `gemini-3-pro-preview` | 0.96 | 0.19 | 0.40 | **0.52** |
| Static | `gpt-5.1` | 0.28 | 0.06 | 0.17 | 0.17 |
| EvoTest | `gpt-5.1` | **0.98** | **0.24** | **0.48** | **0.57** |

## M  DYNAMIC HYPERPARAMETER EVOLUTION: A CASE STUDY

Figure 9 illustrates how the Evolver intelligently manages the Actor's LLM decoding temperature in response to its performance. The Evolver's logic is not random; it follows an adaptive strategy to balance exploration and exploitation.

## N  SCALABILITY WITH MORE CAPABLE MODELS

To investigate how our framework scales with model improvements, we evaluated EvoTest and the non-learning Static baseline using two next-generation models: `openai/gpt-5.1` and `google/gemini-3-pro-preview`. The results, presented in Table 10, show a positive trend: as the underlying model's capability improves, the performance of both agents increases. As models become more capable, EvoTest has more raw material to work with, leading to disproportionately larger gains compared to a simple zero-shot approach.

