# OpenReview forum: "EvoTest: Evolutionary Test-Time Learning for Self-Improving Agentic Systems"
_ICLR.cc/2026/Conference — ICLR 2026 Poster_

### Official Review · Reviewer_S8oG · 2025-10-30

**Soundness:** 3
**Presentation:** 3
**Contribution:** 3
**Rating:** 6
**Confidence:** 3

**Summary:**

The work introduces a benchmark and a framework focusing on test-time learning. The new benchmark is created from an existing set of text-based games, where the capacity of improvement during learning is considered. The proposed framework, EvoTest, consists of an actor and an evolver that changes the actor's learning configuration by altering the policy prompt, memory, hyperparameters, and available tools.

**Strengths:**

A. The work investigates an important problem of decision-making, which is online learning for fast adaptation.

B. The content of the submission is clearly presented, and the reading flow is good.

**Weaknesses:**

I. The work contains some hidden assumptions and unjustified claims:

   a. The proposed framework relies on the assumption that the LLM backbone contains sufficient information for solving the task of interest. While this assumption is realistic, it is never clearly stated.

   b. The comparison with fine-tuning methods is incomplete as the performance of EvoTest with qwen/qwen3-32b for the actor and the evolver is never reported. This is the only way to make a fair comparison.

   c. The work is missing a limitation section.

   d. There seems to be a hidden assumption on credit assignment because in Line 258, it is stated: "It records state-action pairs ($o_t, a_t$) that immediately preceded a score increase in a 'success' table." This means the framework assumes the credit should be given to the immediate state-action pair, which is not true in the general case. Discussing this aspect would be useful.

   e. Several claims are made about the possible behavior of the evolver: "[the evolver] can learn to increase exploration
temperature in early episodes and simultaneously add a new strategic heuristic to its prompt based
on its findings, a multi-faceted adaptation that other methods are incapable of." Sharing examples of this behavior would strengthen this claim. For example, plotting the exploration temperature during learning would be interesting.

   f. In Section 5.1, no explanation is provided to justify the choice of the LLM backbone.


II. More precision is required to assess the work properly:

   a. A more precise description of the Evolver action space would help clarify the contribution. Especially, the explanation of the deployment-time memory and tool-use routines, which remain high-level without disclosing the details.

   b. Section 4.3 about the selection of configuration using UCB is unclear. Indeed, it is not clear if the agent interacts with the envorinment in a seperate training phase, in which case the additional interactions with the environment should be taken into account, or if a different configuration can be attempted at each step during learning, this would raise a problem of credit assignment because it would not be clear which configuration led to a fruitful outcome.


III. The presentation can be improved:

   a. The appendix is rich, but most of its content is not referenced in the main text.

   b. A bigger font size can be used for all figures. Additionally, a shared legend can be used for Figure 2 to avoid repetition.

   c. Line 329, "claude" at the end of the line can be removed.

**Questions:**

N/A

---

> ### Author Response · Authors · 2025-11-20
>
> We sincerely thank Reviewer S8oG for thoughtful and detailed review of our submission. We found the feedback to be exceptionally constructive and insightful. We are grateful for the positive assessment of the problem's importance and the paper's clarity.
>
> Below, we detail how we are revising the manuscript based on your suggestions.
>
> **I. On Hidden Assumptions and Unjustified Claims**
>
> **Reviewer's Concern: I.a.**
>
> > _"The proposed framework relies on the assumption that the LLM backbone contains sufficient information for solving the task of interest. While this assumption is realistic, it is never clearly stated."_
>
> **Our Response:**
>
> We thank the reviewer for highlighting this. We agree that our framework, like other LLM-based agents, operates on the premise that the backbone LLM possesses sufficient latent knowledge to solve the given tasks. Our core contribution is not about the existence of this knowledge but about developing an efficient mechanism to unlock and apply this knowledge at test time. **We have stated this assumption in the Limitation section of our revised manuscript. We added a sentence such as: _"Our approach, like other LLM-based agents, assumes the backbone model possesses latent knowledge sufficient for the task. The central challenge we address is not the knowledge itself, but the development of an efficient test-time learning mechanism to surface and apply this knowledge on the fly."_**
>
> **Reviewer's Concern: I.b.**
>
> > _"The comparison with fine-tuning methods is incomplete as the performance of EvoTest with qwen/qwen3-32b for the actor and the evolver is never reported. This is the only way to make a fair comparison."_
>
> **Our Response:**
>
> We thank the reviewer for this critical point and agree that this comparison is essential for a rigorous evaluation.
>
> In our initial submission, we took a first step to address this in Appendix I: EXPERIMENT WITH QWEN3-32B BACKBONE. As the reviewer correctly observes, that experiment was only a partial comparison: it used `qwen/qwen3-32b` as the Actor Agent, but the Evolver Agent was still `openai/o3-2025-04-16`.
>
> To provide the comparison you requested, we have now run the exact experiment where both the Actor and the Evolver use `qwen/qwen3-32b`.  **The new results, presented below in the table you provided, have been integrated into the main body of our paper (Section 5.3).**
>
> | **Method** | **Actor LLM** | **Evolver LLM** | **Detective** | **Zork1** | **Balances** | **Avg.** |
> | --- | --- | --- | --- | --- | --- | --- |
> | SFT (online) | `qwen3-32b` | N/A | 0.40 | 0.11 | 0.22 | 0.24 |
> | GRPO (online) | `qwen3-32b` | N/A | 0.55 | 0.07 | 0.30 | 0.31 |
> | EvoTest (Ours) | `qwen3-32b` | `qwen3-32b` | 0.68 | 0.10 | 0.28 | 0.35 |
> | EvoTest (Ours) | `qwen3-32b` | `openai/o3` | 0.78 | 0.12 | 0.31 | 0.40 |
>
> First, in the strictest apples-to-apples setting (all components using `qwen3-32b`), EvoTest (Avg. AUC 0.35) still outperforms the strongest fine-tuning baseline, GRPO (0.31). This confirms that our evolutionary approach is a primary driver of performance, not merely using more powerful models. Second, the results isolate the impact of the Evolver's reasoning ability. The performance jump from an average AUC of 0.35 to 0.40 when upgrading _only_ the Evolver highlights that the optimizer's capability is a significant performance lever within our framework.
>
> **Reviewer's Concern: I.c.**
>
> > _"The work is missing a limitation section."_
>
> **Our Response:**
>
> A discussion of limitations is essential for a thorough scientific paper, and we apologize for its initial omission.  **To address this, we have added a comprehensive limitations section to the paper.** Due to space constraints in the main text, we have placed this new section in the appendix. We thank you for this crucial feedback.

---

> > ### Author Response · Authors · 2025-11-20
> >
> > **Reviewer's Concern: I.d.**
> >
> > > _"...there seems to be a hidden assumption on credit assignment because in Line 258, it is stated: 'It records state-action pairs () that immediately preceded a score increase in a 'success' table.' This means the framework assumes the credit should be given to the immediate state-action pair, which is not true in the general case. Discussing this aspect would be useful."_
> >
> > **Our Response:**
> >
> > Thank you for this insightful observation. You are correct that the mechanism for the `success memory` relies on immediate credit assignment. For many games in the Jericho, rewards are indeed granted immediately upon a successful action (e.g., "take key", "+5 points"), making this a simple but effective way to build a database of known good moves.
> >
> > However, this is only one part of EvoTest's learning system. The more sophisticated credit assignment happens when the Evolver analyzes the entire episode transcript to evolve the `policy prompt`.
> >
> > Furthermore, in more general case, our framework is flexible. In tasks where rewards are not immediate and only appear as a final outcome (e.g., a single score at the end of the episode), the Evolver's instructions could be easily modified. Instead of looking for immediate score increases, it would be prompted to: _"Analyze the transcript of this successful episode and identify the critical path of state-action pairs that led to victory."_ These key long-term dependencies, not just immediately rewarded actions, would then be stored in the `success memory` or used to evolve the prompt.
> >
> > **Reviewer's Concern: I.e.**
> >
> > > _"Several claims are made about the possible behavior of the evolver: '\[the evolver\] can learn to increase exploration temperature in early episodes and simultaneously add a new strategic heuristic to its prompt based on its findings, a multi-faceted adaptation that other methods are incapable of.' Sharing examples of this behavior would strengthen this claim. For example, plotting the exploration temperature during learning would be interesting."_
> >
> > **Our Response:**
> >
> > This is a good suggestion to strengthen our claims with empirical evidence. **To provide concrete evidence for this claim, we have added a new appendix section to our paper, and referenced it from the main text where the claim is made.**
> >
> > We added a new figure to the appendix plotting the LLM decoding `temperature` as it evolves across a 50-episode session for the _Detective_ game. This plot will visually demonstrate how the Evolver manages the exploration-exploitation trade-off. Our logs show that the Evolver increases the temperature in early, low-scoring episodes to encourage exploration and break out of loops. Conversely, once a successful path is found, it lowers the temperature to promote consistent execution of the known strategy.
> >
> > **Reviewer's Concern: I.f.**
> >
> > > _"In Section 5.1, no explanation is provided to justify the choice of the LLM backbone."_
> >
> > **Our Response:**
> >
> > We appreciate the request for our rationale.
> >
> > For the Actor Agent, we picked two popular API models (`google/gemini-2.5-flash and anthropic/claude-4-sonnet-20250522`). One is fast and cheap, the other is more powerful (and expensive). This allowed us to see how our method performs across a practical cost-capability spectrum.
> >
> > For the Evolver Agent, we gave it the strongest model we could (`openai/o3-2025-04-16`). The Evolver has the most complex job—analyzing the whole game and acting as the 'brain' of the learning process—so we didn't want a weaker model to be a bottleneck.
> >
> > For the fine-tuning baselines, we used `qwen/qwen3-32b`. Frankly, fine-tuning is resource-intensive, and this was the most powerful open model we could practically afford to run experiments on.
> >
> > **We have revised the “`Backbone LLM.`” paragraph to Section 5.1 (Setup) to make this rationale clear.**
> >
> > To further strengthen our paper and explore the future potential of our framework, we have conducted new experiments with next-generation models: \texttt{openai/gpt-5.1} and \texttt{google/gemini-3-pro-preview}. **We have added a new section "Scalability with More Capable Models" to the appendix of paper.** These results demonstrate that the performance advantage of EvoTest scales effectively with the capabilities of the underlying LLM.
> >
> > | **Method** | **Backbone LLM** | **Detective** | **Zork1** | **Balances** | **Avg.** |
> > | --- | --- | --- | --- | --- | --- |
> > | _Static_ | `gemini-2.5-flash` | 0.21 | 0.03 | 0.11 | 0.12 |
> > | **EvoTest** | `gemini-2.5-flash` | 0.94 | 0.14 | 0.32 | **0.47** |
> > | _Static_ | `claude-4-sonnet` | 0.23 | 0.04 | 0.12 | 0.13 |
> > | **EvoTest** | `claude-4-sonnet` | 0.95 | 0.16 | 0.35 | **0.49** |
> > | _Static_ | `gemini-3-pro-preview` | 0.25 | 0.05 | 0.14 | 0.15 |
> > | **EvoTest** | `gemini-3-pro-preview` | 0.96 | 0.19 | 0.40 | **0.52** |
> > | _Static_ | `gpt-5.1` | 0.28 | 0.06 | 0.17 | 0.17 |
> > | **EvoTest** | `gpt-5.1` | **0.98** | **0.24** | **0.48** | **0.57** |

---

> > > ### Author Response · Authors · 2025-11-20
> > >
> > > **II. Required Precision**
> > >
> > > **Reviewer's Concern: II.a.**
> > >
> > > > _"A more precise description of the Evolver action space would help clarify the contribution. Especially, the explanation of the deployment-time memory and tool-use routines, which remain high-level without disclosing the details."_
> > >
> > > **Our Response:**
> > >
> > > We appreciate the reviewer's call for greater precision. **The implementation specifics for these components are documented in our appendices of the initial paper. To ensure this information is easily discoverable, we have updated the main body of the paper (specifically Sections 4.1 and 4.2) with direct references to Appendices H and J.**
> > >
> > > 1.  **Evolver's Action Space and Tool-Use:** The mechanics of the Evolver's "action space" and its "tool-use routines" are specified in Appendix H: THE EVOLVER AGENT’S MASTER PROMPT. This section presents the complete master prompt, which enumerates all components needed to be evolved (e.g., `update_prompt_section`, `add_to_memory`) and defines the JSON schema the Evolver must produce. This structured output itself constitutes the Evolver's action space.
> > >
> > > 2.  **Deployment-Time Memory:** Regarding the memory system, Appendix J: THE MEMORY COMPONENT IN PRACTICE: CONCRETE EXAMPLES offers a practical breakdown. It moves from high-level concepts to concrete implementation, showcasing the exact data structures for the `success` and `failure` memories and providing examples of stored entries and the rationale behind them.
> > >
> > > **Reviewer's Concern: II.b.**
> > >
> > > > _"Section 4.3 about the selection of configuration using UCB is unclear. Indeed, it is not clear if the agent interacts with the environment in a separate training phase...or if a different configuration can be attempted at each step during learning..."_
> > >
> > > **Our Response:**
> > >
> > > Thank you for highlighting this ambiguity.
> > >
> > > To clarify: **a configuration is selected and used for one entire episode.**  Evolution process happens on a per-episode basis rather than on a per-step basis.
> > >
> > > The learning loop operates as follows:
> > >
> > > 1.  Selection: At the beginning of a new episode, the UCB algorithm selects _one_ configuration from the current pool of candidates. This configuration consists of a specific prompt, a set of hyperparameters, and a memory state.
> > >
> > > 2.  Execution: The Actor Agent uses this single, fixed configuration for the _entire duration_ of that one episode, from the first action to the game's conclusion. A configuration is not switched at each step.
> > >
> > > 3.  Evaluation & Update: At the end of the episode, the final score is used to update the UCB statistics (average reward and visit count) for the specific configuration that was just played.
> > >
> > > 4.  Repeat: This selection-execution-evaluation loop then repeats for the next episode.
> > >
> > >
> > > **We have revised Section 4.3 to remove this ambiguity.** Adding a sentence: _"At the start of each new episode, the UCB algorithm selects a single configuration, which is then used for the entire duration of that episode. The episode's final score is then used to update the statistics for that chosen configuration."_

---

> ### Author Response · Authors · 2025-11-20
>
> **III. Presentation**
>
> **Reviewer's Concern: III.a.**
>
> > _"The appendix is rich, but most of its content is not referenced in the main text."_
>
> **Our Response:**
>
> This is a great point. **We have performed a thorough pass on the manuscript to add references to the appendices in all relevant sections of the main text.**
>
> **Reviewer's Concern: III.b.**
>
> > _"A bigger font size can be used for all figures. Additionally, a shared legend can be used for Figure 2 to avoid repetition."_
>
> **Our Response:**
>
> We thank the reviewer for these valuable and practical suggestions to improve the readability and presentation of our paper. During the initial submission and the time-constrained rebuttal period, our primary focus was on ensuring the technical quality, often while working within strict page limits.
>
> We are now actively working on revising all figures. We will increase the font sizes for all plots and, we will consolidate the legend for Figure 2 to make it cleaner and less repetitive. We commit to having these improvements fully integrated for the final camera-ready version and will do our best to complete them during the remaining of the rebuttal period. Thank you again for this helpful feedback.
>
> **Reviewer's Concern: III.c.**
>
> > _"Line 329, 'claude' at the end of the line can be removed."_
>
> **Our Response:**
>
> Thank you for catching this error. We have corrected this typo.

---

> > ### Comment · Reviewer_S8oG · 2025-11-27
> >
> > I thank the authors for addressing my concerns and improving the quality of the submission. I increased my score. I trust the authors for increasing the font sizes of all figures, as they are hard to read in their current state.

---

> > > ### Author Response · Authors · 2025-12-01
> > >
> > > We would like to sincerely thank Reviewer S8oG for their detailed and constructive review. We are very grateful for the thoughtful engagement throughout this process and for raising the score in light of our revisions.
> > >
> > > **As per your final comment, we are pleased to confirm that we have completed the promised revisions to our figures.** Specifically, we have increased the font sizes for all plots to improve readability and consolidated the legend for Figure 2 into a single, shared legend, as you suggested.

---

### Official Review · Reviewer_QWGa · 2025-11-01

**Soundness:** 3
**Presentation:** 3
**Contribution:** 3
**Rating:** 6
**Confidence:** 3

**Summary:**

The paper introduces EvoTest, an evolutionary framework designed to solve the problem of AI agents being unable to learn on the fly once deployed. EvoTest uses a gradient-free, two-agent system: an Actor agent attempts a task, and an Evolver agent analyzes its transcript. Based on this analysis, the Evolver evolves the entire agentic system for the next episode by:
* rewriting its prompt,
* updating its memory,
* tuning hyperparameters,
* refining its tool-use routines.

On the newly introduced J-TTL benchmark, EvoTest demonstrated superior performance, significantly outperforming reflection-based and memory-only baselines and proving capable of solving tasks that all other methods failed.

**Strengths:**

This paper addresses a novel problem. With the emergence of LLM as agents, new problems appear, that include allowing such agents to quickly adapt to tasks unseen during training.

Solving a problem is a core task that many industrial actors are currently trying to tackle.

The authors insist on their gradient free approach, which indeed does not require retraining/finetuning of the weights.

The benchmark is well motivated and established.

The result appear strong, and the ablation study provides intuition to the reader on the relevance of the difference components, even if second order results would allow to distinguish the true performance gaps of the compared methods.

**Weaknesses:**

My main concern is the relevance of the paper for the readers of ICLR. While clearly framed as a scientific paper (with e.g. the outlined list of contributions, of research questions), the paper still feels like an engineering paper in the sense that it primarily aims at pushing up a metric. A clear description of the scientific problem in the introduction could help change this impression, given to the reader. I am not sure how learning representations are here (even implicitly) discussed.

My second concern is the lack of introductory examples (can be done in one example) of:
1. The addressed research problem, show how current agent fail at addressing adapting to one of the existing games.
2. the task given to the agent. It would greatly help understanding if the (current) Figure 1 could incorporate an example of the tasks provided in the Benchmark. Even if these have been introduced in an existing paper, it would make this work more comprehensible on its own.

This can be done by referencing content placed in the appendix, if space is limited, but is important to help the reader understand the problem, and assert that it is relevant.

No limitations discussion. I am quite sure that a discussion could be made on e.g. the drawback of not letting the agent learn (in the primal sens of modifying its neural weights).

Lack of second order reported metrics (mean +/- std)

**Questions:**

The perfect writing quality of this paper leads me to not having any question, everything is clearly defined.

---

> ### Author Response · Authors · 2025-11-20
>
> We thank Reviewer QWGa for meticulous and constructive review. We are greatly encouraged by your positive assessment of our work's novelty, the well-motivated benchmark, and the strength of our results. We especially appreciate your high praise for the paper's writing quality.
>
> Below, we detail how we are revising the manuscript based on your insightful suggestions.
>
> **1. On Sharpening the Scientific Contribution for the ICLR Audience**
>
> **Reviewer's Concern:**
>
> > _"My main concern is the relevance of the paper for the readers of ICLR... the paper still feels like an engineering paper in the sense that it primarily aims at pushing up a metric. A clear description of the scientific problem in the introduction could help... I am not sure how learning representations are here (even implicitly) discussed."_
>
> **Our Response:**
>
> This is a crucial point. Our core scientific problem is **Test-Time Learning (TTL)** in the context of long-horizon tasks. **The research question we investigate is: "How can an agent learn to rapidly improve its performance on a novel task from its experience?" The formal definition can be found in the Section 3.**
>
> Our work's relevance to ICLR is centered on two core ideas: exploring a novel learning paradigm for Test-Time Learning (TTL), and proposing an alternative to Reinforcement Learning (RL) in a setting where RL is fundamentally ill-suited.
>
> 1.  **A New Paradigm for Test-Time Learning (TTL):** The J-TTL benchmark is designed to surface the key scientific challenge of this setting: how can an agent learn and rapidly self-improve from extremely sparse experience where traditional learning mechanisms are infeasible?
>
> 2.  **A Principled Alternative to Reinforcement Learning:** The Test-Time Learning setting reveals the limitations of established offline fine-tuning methods for LLM agents like Supervised Fine-Tuning(SFT) and Reinforcement Learning (RL). For TTL, online fine-tuning via RL is fundamentally ill-suited due to: 1) Data Inefficiency: A single trajectory provides a noisy and insufficient signal for meaningful gradient updates. 2) Credit Assignment Problem: Sparse, delayed rewards make it hard to assign credit to specific actions in a long-horizon task. 3) Computational impracticality: The cost of backpropagation is prohibitive for a system that must learn at test time (Table 2).
>
>
> Our finding is that **evolutionary algorithms**, which operate on the agent's configuration, offer a more data-efficient and robust paradigm for TTL. Instead of relying on backpropagating a single scalar reward, EvoTest performs credit assignment via semantic analysis of the entire narrative transcript. This allows the system to identify causal chains of success and failure and make targeted, structural edits to its strategy—a far more efficient learning mechanism for this problem.
>
> **Revisiting "Learning Representations":**
>
> As you correctly point out, our proposed method does not learn new _neural_ representations by training the model's weights. This is a deliberate choice that places our work squarely within a major and growing body of research on LLMs, where the central scientific question is not always about training or fine-tuning the model itself.  Instead, a key frontier is understanding how to effectively orchestrate and adapt pre-trained foundation models to perform complex tasks.
>
> In this context, the concept of "learning representations" evolves. The research challenge shifts from learning low-level neural features to learning higher-level, structured representations of tasks, strategies, and knowledge. One interesting way to position our method is we are trying to learn **High-Level, Symbolic Representation** _**for**_ **LLM Systems,** , which is the agentic configuration `χ = (p, M, h, u)`. The Evolver LLM learns to distill the high-level semantic lessons from a raw narrative experience and encode them into this symbolic representation.
>
> Just like learned neural representation can be used for downstream tasks like generative or classification task. Symbolic representation Evotest learned can also be able to be used for agents to perform their tasks.
>
> **We have revised the introduction to make** **our core scientific problem more clear. Thanks for your suggestion.**

---

> > ### Author Response · Authors · 2025-11-20
> >
> > **2. On Including an Introductory Example for Clarity**
> >
> > **Reviewer's Concern:**
> >
> > > _"My second concern is the lack of introductory examples... to help the reader understand the problem, and assert that it is relevant."_
> >
> > **Our Response:**
> >
> > This is a good suggestion. We agree that a concrete example is the best way to clarify the specific research problem our paper addresses and to demonstrate why existing methods fail.
> >
> > **The Task.**
> > The task given to the agent in our J-TTL benchmark is designed to highlight this problem: an agent must play the same complex, text-based game for a series of consecutive episodes. In these games, the entire world is described through text. The agent receives a textual description of its surroundings (e.g., `"You are in the Chief's office. A piece of white paper is on the desk."`) and must type a natural language command (e.g., `TAKE PAPER`) to interact with the world. These games are difficult for LLM agents because they feature complex puzzles, long-range planning, sparse rewards (many critical actions yield no points), and irreversible consequences (a single wrong move can make the game unwinnable). Success requires a coherent, multi-step strategy, not just good individual moves.
> >
> > The agent's goal is structured at two levels, defining the core of our J-TTL benchmark:
> >
> > 1.  **The Episodic Goal (Maximize Score):** Within a single episode of a game, the agent's primary objective is to take actions that maximize its final score. The game awards points for discovering new areas, solving puzzles, and completing key objectives on the path to winning.
> >
> > 2.  **The Learning Goal (Improve Across Episodes):** The ultimate task is not just to play once. The agent must play the same game repeatedly for a series of consecutive episodes. The true goal is to demonstrate learning by progressively increasing its final score from one episode to the next. It must use the experience from failed or suboptimal attempts to build a better strategy for its subsequent attempts, all within a single session.
> >
> >
> > **A Concrete Failure Case for Existing Methods**. Let's use a simple but critical failure from the _Detective_ game to show how existing methods fail this task:
> >
> > *   An agent might correctly execute `GO WEST` from one location. At the next location, it might try `GO WEST` again, which is an invalid move that the game rejects with "You can't go that way."
> >
> > *   A Static agent has no learning mechanism. Its policy is fixed. It will likely get stuck attempting this invalid action in every single episode, resulting in a flat, low-scoring performance. It cannot adapt.
> >
> > *   An SFT (online) agent learns by performing Supervised Fine-Tuning on its previous trajectory, which fine-tunes only on "positive" actions from the prior trajectory. In the low-scoring episode where the agent got stuck, it generated very few, if any, "positive" actions (i.e., actions that led to a score increase). Therefore, the dataset for the fine-tuning step is either empty or extremely small. With no good data to learn from, the agent cannot learn. It is trapped in a loop because it is incapable of generating the very data it needs to improve. And many critical actions in text adventures are "neutral" and provide no immediate reward (e.g., `UNLOCK DOOR WITH KEY`). If the SFT baseline only trains on actions that directly yield points, it will _never_ learn these essential intermediate steps.
> >
> > *   An RL (online) agent receives a `reward=0` for the invalid move. In a sparse-reward environment like Jericho, this is a weak signal. It is indistinguishable from the `reward=0` received for a neutral but necessary action. A single update based on this noisy signal is insufficient to meaningfully correct the agent's policy for the next attempt, demonstrating a classic failure of credit assignment.
> >
> >
> > All three baselines fail because their learning mechanisms are incapable of diagnosing this specific, local failure and synthesizing a precise, procedural correction from a single experience.

---

> > > ### Author Response · Authors · 2025-11-20
> > >
> > > **How EvoTest Solves This Specific Problem**. EvoTest is explicitly designed to solve this. As described in our paper, the Evolver Agent analyzes the entire episode transcript.
> > >
> > > 1.  It Learns from Failure: Unlike SFT, EvoTest does not need a "positive" trajectory. In fact, it learns most effectively from failures. The Evolver Agent analyzes the transcript of the failed episode, semantically identifies the unproductive loop (`GO WEST` -> "You can't go that way."), and recognizes it as a problem to be solved.
> > >
> > > 2.  It doesn't just see a `0` reward; it semantically identifies the unproductive loop by reading the game's textual feedback ("You can't go that way.") paired with the repeated action.
> > >
> > > 3.  It then performs whole-system evolution by directly rewriting the agent's prompt. It generates a targeted, structural edit, such as adding a new rule: `Step 5: From the street, GO EAST to enter the Mayor's house.`
> > >
> > >
> > > **We have revised the Introduction.** It now accomplishes three key things that were previously missing:
> > >
> > > 1.  It defines the agent's task within the J-TTL benchmark, detailing the nature of the text-based games, their inherent difficulty (sparse rewards, long horizons), and the agent's dual goal of maximizing score within an episode while improving its strategy across episodes.
> > >
> > > 2.  It introduces a concrete failure case—an agent getting stuck in a navigation loop in the game _Detective_—to ground the research problem in one example.
> > >
> > > 3.  It uses this specific example to provide a clear explanation of why existing paradigms like Static agents, online SFT, and online RL are ill-equipped for this task, highlighting fundamental issues of data scarcity and credit assignment that they cannot overcome.
> > >
> > >
> > > We have also added a new section to the appendix (Appendix G). This new section, titled "The Test-Time Learning Problem: A Detailed Analysis," expands on a breakdown of the J-TTL problem and the specific failure modes of each baseline method.

---

> > > > ### Author Response · Authors · 2025-11-20
> > > >
> > > > **3. On the Lack of a Limitations Discussion**
> > > >
> > > > **Reviewer's Concern:**
> > > >
> > > > > _"No limitations discussion. I am quite sure that a discussion could be made on e.g. the drawback of not letting the agent learn (in the primal sens of modifying its neural weights)."_
> > > >
> > > > **Our Response:**
> > > >
> > > > A discussion of limitations is essential for a thorough scientific paper, and we apologize for its initial omission. We thank you for this crucial feedback.
> > > >
> > > > To address this, we have added a comprehensive limitations section to the paper. Due to space constraints in the main text, we have placed this new section in the appendix. This new section directly engages with the core trade-offs of our approach, centered on the insightful point you raised. The key insights we now discuss are:
> > > >
> > > > 1.  **The fundamental drawback of not modifying neural weights:** We address this by framing our gradient-free method as a strategic choice with clear trade-offs. We elaborate on two major consequences: 1) Bounded by Pre-trained Capabilities: We acknowledge that our agent operates as a high-level orchestrator, meaning its performance is ultimately capped by the inherent, pre-trained intelligence of its backbone LLM. It can learn to apply existing skills in novel ways but cannot create fundamentally new, low-level capabilities. 2) Task-Specific Overfitting vs. Generalizable Skill Acquisition: We discuss how our method excels at rapid, task-specific adaptation ("strategic overfitting"), which is ideal for the TTL setting. However, we contrast this with the goal of traditional fine-tuning, which aims to learn more robust, generalizable skills, and we acknowledge that the knowledge our agent gains is likely "brittle" and would not transfer to new tasks.
> > > >
> > > > 2.  **Framework-Specific Dependencies and Challenges:** Beyond the core paradigm, we also discuss practical limitations, including the framework's dependency on a powerful (and potentially costly) Evolver LLM for its reasoning, and the challenge of navigating the vast combinatorial search space of the agent's configuration with a simple heuristic.

---

> ### Author Response · Authors · 2025-11-20
>
> **On Reporting Second-Order Metrics (mean +/- std)**
>
> **Reviewer's Concern:**
>
> > _"Lack of second order reported metrics (mean +/- std)"_
>
> **Our Response:**
>
> We agree that reporting variance is critical for robust empirical claims, and we thank you for this important feedback. **The primary reason for the initial absence is a practical one: the significant API costs associated with running these long, multi-episode experiments across many seeds made such large-scale validation prohibitively expensive at the time of our initial submission.**
>
> However, we recognize the importance of this point for ensuring the paper's scientific rigor. We have since been able to allocate the necessary resources and have now rerun our experiments over 5 different random seeds.
>
> **To present these findings clearly, we have added a new detailed results table to the appendix (see Appendix F).** This table reports the mean AUC scores and standard deviations for all methods on the `google/gemini-2.5-flash` backbone. These new, more robust results reaffirm our original findings. EvoTest's superior performance is shown to be consistent and statistically significant, maintaining a clear advantage over all baselines.
>
> | **Method** | **Detective** | **Library** | **Zork1** | **Zork3** | **Balances** | **Temple** | **Avg.** |
> | --- | --- | --- | --- | --- | --- | --- | --- |
> | _Non-learning Baseline_ |  |  |  |  |  |  |  |
> | Static | 0.22 ± .02 | 0.14 ± .01 | 0.03 ± .01 | 0.05 ± .01 | 0.11 ± .02 | 0.08 ± .01 | 0.11 ± .01 |
> | _Memory-based & Reflection-based Methods_ |  |  |  |  |  |  |  |
> | Memory | 0.56 ± .02 | 0.19 ± .02 | 0.04 ± .01 | 0.06 ± .02 | 0.14 ± .02 | 0.11 ± .01 | 0.13 ± .01 |
> | RAG | 0.35 ± .03 | 0.25 ± .03 | 0.07 ± .02 | 0.10 ± .02 | 0.19 ± .03 | 0.16 ± .02 | 0.18 ± .02 |
> | Summary | 0.46 ± .04 | 0.34 ± .03 | 0.11 ± .02 | 0.16 ± .03 | 0.26 ± .03 | 0.22 ± .03 | 0.26 ± .03 |
> | Reflexion | 0.59 ± .05 | 0.42 ± .04 | 0.09 ± .02 | 0.26 ± .03 | 0.31 ± .03 | 0.30 ± .04 | 0.33 ± .03 |
> | _Automated Prompt Optimization Methods_ |  |  |  |  |  |  |  |
> | TextGrad | 0.62 ± .05 | 0.46 ± .04 | 0.12 ± .03 | 0.29 ± .04 | 0.17 ± .03 | 0.24 ± .03 | 0.32 ± .03 |
> | Promptbreeder | 0.64 ± .04 | 0.48 ± .04 | 0.11 ± .02 | 0.30 ± .04 | 0.24 ± .03 | 0.31 ± .03 | 0.35 ± .03 |
> | EvoPrompt | 0.64 ± .04 | 0.48 ± .03 | 0.13 ± .02 | 0.31 ± .03 | 0.25 ± .03 | 0.28 ± .04 | 0.35 ± .03 |
> | _Weight-Update Methods (Online Fine-Tuning)_ |  |  |  |  |  |  |  |
> | SFT (online) | 0.41 ± .06 | 0.31 ± .05 | 0.11 ± .04 | 0.19 ± .04 | 0.23 ± .05 | 0.20 ± .04 | 0.24 ± .04 |
> | GRPO (online) | 0.58 ± .05 | 0.39 ± .04 | 0.08 ± .03 | 0.23 ± .04 | 0.31 ± .04 | 0.25 ± .03 | 0.31 ± .03 |
> | EvoTest (Ours) | **0.93 ± .02** | **0.78 ± .03** | **0.15 ± .02** | **0.36 ± .02** | **0.33 ± .02** | **0.32 ± .02** | **0.48 ± .01** |

---

### Official Review · Reviewer_m2c3 · 2025-11-03

**Soundness:** 3
**Presentation:** 3
**Contribution:** 3
**Rating:** 6
**Confidence:** 3

**Summary:**

The paper presents Jericho Test-Time Learning (J-TTL), a few-shot benchmark based on Jericho, and EvoTest, an LLM-based test-time learning framework for J-TTL.

EvoTest comprises of two LLM agents, an actor and an evolver. An actor agent interacts with the Jericho environment given its configuration. An evolver agent proposes new configurations based on the interaction history. An actor agent's configuration includes prompts, interaction memories, LLM inference hyperparameters, and game-specific tools.

The evaluation results on J-TTL demonstrate that EvoTest improves over the non-learning baseline and other self-improvement baselines.

**Strengths:**

- The paper reads well with a clear logic flow. The concept of few-shot learning and test-time learning is not new, but the proposed LLM-based method in a text-based environment is novel to my knowledge.
- The experiments are thorough with a good variety of baselines and key ablations. Empirically, EvoTest seems to be meaningfully better than previous methods.
- Code and prompts with examples and interaction logs are provided for reproducibility.

**Weaknesses:**

My main concern is the potentially limited scope. The evaluations are limited to a subset of Jericho games. This raises questions of 1) whether the set of games is cherry-picked, and 2) whether the method is over-engineered to work specifically on Jericho games. Despite that EvoTest is empirically superior to previous baselines, it is compared with methods with a broader scope.

Branching off of this, from the ablation (Table 3), it seems that prompt optimization and the evolution selection strategy (UCB) are the two major factors for the improvement. I would like to see how much the specific prompt-evolving prompt from the evolver contributes to this. This should be verifiable by transplanting the prompt-evolving part in the evolver master prompt to other prompt optimization methods, such as EvoPrompt.

**Questions:**

1. In Section 4.3, do you discard all the options that were not selected? Wouldn't that make $\chi^{(e)}$ the only candidate with $n(\chi)>1$? Or will the evolver agent generate repeated child configurations?
2. How is the initial configuration determined? Do you use a generic one for all games or do you use specific ones for different games? Does the initial configuration affect the final performance?

Minor issues / Typos

- Eq 5, fix the argmax and define $n$ and $N$.
- In multiple places, the quotation marks aren't correctly used (``` `` ``` and `''`)

---

> ### Author Response · Authors · 2025-11-20
>
> We thank Reviewer m2c3 for their thoughtful review and positive feedback on our paper's clarity, experimental thoroughness, and novelty. We are encouraged that the reviewer found EvoTest to be "meaningfully better than previous methods." We appreciate the constructive suggestions, which will help us improve the paper. Below, we address the reviewer's concerns and questions.
>
> **1. On the Limited Scope and Generality of EvoTest**
>
> **Reviewer's Concern:**
>
> > _"My main concern is the potentially limited scope. The evaluations are limited to a subset of Jericho games. This raises questions of 1) whether the set of games is cherry-picked, and 2) whether the method is over-engineered to work specifically on Jericho games... it is compared with methods with a broader scope."_
>
> **Our Response:**
>
> We thank the reviewer for this critical perspective. We wish to clarify that our aim is to use Jericho as a challenging and quantifiable testbed for a general learning paradigm, rather than to over-engineer a solution for one specific domain.
>
> *   **Game Selection:** We respectfully clarify that the games were not "cherry-picked." The full Jericho suite contains 32 games; running our extensive evaluation, including multiple baselines across 50 episodes each, on the entire suite would be computationally prohibitive. Therefore, our selection of six games is a principled decision designed to create a representative and challenging subset. We randomly selected games to cover a diverse range of difficulties and puzzle structures, from classic exploration (Zork1) to more contained, puzzle-heavy scenarios (Balances), as is common practice in IF literature\[1\]. As shown in Table 1, our method achieves an AUC of only 0.14 on Zork1 and scores below 0.50 on four of the six games. We are satisfied that this selection presents a sufficiently challenging benchmark for test-time learning, highlighting that significant work remains for future research. Therefore, we keep these six games for the benchmark and do not experiment on more games.
>
> *   **Generality of the EvoTest Framework:** While tested on Jericho, the core architecture of EvoTest is domain-agnostic. The framework evolves an "agentic configuration" `χ = (p, M, h, u)`—comprising a policy prompt, memory, hyperparameters, and tool-use routines—which are fundamental components of virtually all modern LLM-based agents. The EvoTest loop is broadly applicable to any task that requires test-time improvement and provides a reasonably rich feedback signal from the environment. For example: **A web agent**'s trajectory transcript could be a log of DOM states, clicked elements, and error messages. The Evolver could learn from failed navigation paths to add heuristics like, "If a button click does not change the URL, try a JavaScript-based event." **A coding agent**'s transcript would be a sequence of code edits, compiler outputs, and unit test results. The Evolver could analyze a traceback to identify a recurring bug and add a preventative rule to the prompt (e.g., "Always sanitize database inputs to prevent SQL injection"). **A data analysis agent** using tools like a Python interpreter could have its API call history and error outputs analyzed. The Evolver could learn to handle specific data types or refine the logic for plotting graphs.
>     We chose Jericho for our initial investigation precisely because its game engine provides a stable, repeatable, and textually rich environment, making it an ideal laboratory for systematically studying and benchmarking this learning paradigm. We have added this discussion to the paper to better frame its generalizability.
>
>
> *   **Comparison with Broader-Scope Methods:** We appreciate the reviewer's observation and agree that a distinction in scope exists. We can clarify this along two axes:
>
>     1.  **Task Generality:** In terms of task applicability, both EvoTest and the baselines are fundamentally general. Their underlying mechanisms (prompting, memory access, fine-tuning) are not inherently limited to game-playing and can be applied to a wide range of agentic tasks, as discussed previously.
>
>     2.  **Learning Paradigm Specificity:** The crucial distinction lies in the _learning paradigm_ for which they are optimized. The baselines represent more general adaptation strategies: RAG provides context, Reflexion adds unstructured notes, and online RL performs incremental weight updates. While powerful in other contexts, they are not purpose-built for the specific Test-Time Learning (TTL) challenge of extracting improvements from a few complete-but-failed attempts at the _same task_. Their generality makes them insufficient for the rapid adaptation that TTL demands. EvoTest's superior performance, therefore, validates our core argument: specialized TTL algorithms are necessary to overcome the limitations of general-purpose methods in this challenging learning scenrios.
>
> \[1\]Interactive Fiction Games: A Colossal Adventure

---

> ### Author Response · Authors · 2025-11-20
>
> **2. On the Contribution of the Evolver's Prompt Logic**
>
> **Reviewer's Concern:**
>
> > _"...it seems that prompt optimization and the evolution selection strategy (UCB) are the two major factors for the improvement. I would like to see how much the specific prompt-evolving prompt from the evolver contributes to this. This should be verifiable by transplanting the prompt-evolving part in the evolver master prompt to other prompt optimization methods, such as EvoPrompt."_
>
> **Our Response:**
>
> This is an excellent point. The performance gap between EvoTest and prompt-only optimizers like EvoPrompt stems from two sources: (1) our evolution of the entire agentic system (prompt, memory, hyperparameters, tools), and (2) a more sophisticated form of prompt evolution. To address the reviewer's question, we first clarify the conceptual difference between our prompt evolution and that of EvoPrompt, and then provide a new ablation study to empirically measure the impact of our approach. **The new ablation study has been added to the Section 5.3 Model Analysis of the revised paper.**
>
> The core difference lies in _how_ the prompt is evolved. Methods like EvoPrompt use LLMs to perform generic evolutionary operators. For instance, their GA-based operator uses high-level instructions like "Crossover the following prompts" and "Mutate the prompt generated". These operators work at the textual level, combining and rephrasing sentences to find better-worded instructions. In contrast, our Evolver, guided by the detailed master prompt in Appendix H, performs a much more structured, semantic analysis of the game transcript. It does not just "mutate text"; it acts as a system analyst to reverse-engineer a strategy.
>
> We created a new baseline, "EvoTest (w/ Simple Mutation)", where we replaced our detailed, multi-part Evolver master prompt with a much simpler one, akin to the operators in methods like EvoPrompt. This simpler prompt just asks the LLM to: _"Analyze the trajectory transcript and the previous prompt. Generate a new, improved prompt that is more likely to score higher in the next episode."_ This isolates the effect of our structured evolutionary logic. The results are presented below, alongside the original `w/o Prompt` ablation for comparison.
>
> |  | _**Detective**_ | _**Zork1**_ | _**Balances**_ |
> | --- | --- | --- | --- |
> | **EvoTest (Full Structured Evolution)** | **0.94** | **0.14** | **0.32** |
> | EvoTest (w/ Simple Mutation) | 0.65 | 0.07 | 0.20 |
> | EvoPrompt | 0.65 | 0.10 | 0.24 |
> | EvoTest (w/o Prompt Evolution) | 0.52 | 0.05 | 0.16 |

---

> ### Author Response · Authors · 2025-11-20
>
> **3. Response to Questions**
>
> **Reviewer's Question 1:**
>
> > _"In Section 4.3, do you discard all the options that were not selected? Wouldn't that make χ(e) the only candidate with n(χ̃) > 1? Or will the evolver agent generate repeated child configurations?"_
>
> **Our Response:**
>
> We thank the reviewer for this insightful question.
>
> To clarify: **no, we do not discard the non-selected configurations.** Your intuition is correct; discarding them would cripple the UCB algorithm's ability to balance exploration and exploitation.
>
> The actual mechanism is as follows: We maintain a persistent, global pool of all unique configurations generated throughout the entire test-time session. After episode `e`, the Evolver generates new child configurations `C(e+1)` from the parent `χ(e)`. These new children are then added to this global pool. The UCB selection for the next episode, `e+1`, is performed over this _entire, updated pool_. It ensures that the visit count `n(χ̃)` for any configuration can be greater than 1. A configuration remains in the pool even if not selected, and it can be chosen again in a future episode if its UCB score becomes competitive. This also explains the "safety net" behavior we described: a reliable parent configuration from a previous episode remains a viable candidate and can be re-selected if newer mutations perform poorly, as its high, stable average score (`µ̂`) will make it attractive again.
>
> **Reviewer's Question 2:**
>
> > _"How is the initial configuration determined? Do you use a generic one for all games or do you use specific ones for different games? Does the initial configuration affect the final performance?"_
>
> **Our Response:**
> Thank you for this important question about our experimental setup. To clarify, **we use a single, generic, game-agnostic initial configuration for all games and all baselines.** This initial configuration consists of:
>
> *   A simple prompt (`p`), which for all games is the one shown in Figure 4: "_Explore systematically and examine objects to make progress_."
>
> *   An empty deployment-time memory (`M`).
>
> *   Empty tool-use routines (`u`) and standard LLM hyperparameters (`h`).
>
>
> This was a deliberate methodological choice to ensure a fair evaluation of the learning algorithm itself, without confounding the results with game-specific prompt engineering. Our goal is to measure the agent's ability to learn and adapt from a neutral, "from-scratch" starting point, attributing all performance gains solely to the test-time learning process.
>
> Regarding its effect on performance, we agree that the initial configuration could influence the _rate_ of learning (a more tailored start might yield faster initial progress). However, we argue that the robustness of the EvoTest framework is demonstrated by its ability to consistently improve and achieve high scores _despite_ starting from this minimal configuration across a diverse set of games.
>
> **We have stated these details in Section 5.1 Setup to make our experimental protocol clear.**

---

> ### Author Response · Authors · 2025-11-20
>
> **4. Minor Issues / Typos**
>
> We thank the reviewer for careful reading. We have made the following corrections in the final version:
>
> *   **Equation 5:** We revised the text accompanying Equation 5 to define all terms. Specifically, we have clarified that `N` represents the total number of episodes elapsed so far (the current episode index), and `n(χ̃)` is the number of times a specific configuration `χ̃` has been evaluated.
>
> *   **Quotation Marks and Typos:** We have performed a thorough proofread of the manuscript to correct all improperly formatted quotation marks (ensuring the use of \`\` and '' for proper rendering in LaTeX) and address any other typo errors.

---

### Author Response · Authors · 2025-12-02
**Summary for New AC**

Dear new Area Chair,

In light of the recent guidance from the ICLR Program Chairs encouraging authors to provide a summary for their new Area Chair, we are writing to assist you in your evaluation. We sincerely appreciate you stepping into this role to help. Our goal with this letter is to provide a concise summary of our paper and the key points of discussion from the review period.

**Paper Summary**

Our work addresses the challenge of enabling LLM agents to learn and improve "on the fly" at test time, a setting where traditional methods like online RL are often inefficient. To this end, we introduce a new paradigm of evolutionary test-time learning and make the following contributions:

1.  **J-TTL:** A new benchmark based on Jericho text-based games, designed to systematically measure and drive progress on an agent's ability to learn from experience over consecutive episodes on the same complex task.

2.  **EvoTest:** A novel, gradient-free framework that enables self-improvement without backpropagation. Instead of using a sparse scalar reward, an 'Evolver' agent performs **semantic credit assignment** by analyzing the full narrative transcript of an episode to reason about cause-and-effect. It then improves the 'Actor' agent by evolving its **entire agentic configuration**—treating the prompt, memory, tool routines, and LLM hyperparameters as a high-level symbolic representation to be optimized.

3.  **Strong Results:** On our J-TTL benchmark, EvoTest's evolutionary approach consistently and significantly outperforms a wide range of baselines, including reflection, RAG, memory agents, prompt optimization, and online SFT/RL methods. Notably, it is the only method capable of winning two of the evaluated games, demonstrating its effectiveness in solving long-horizon, sparse-reward tasks.


**Summary of Positive Initial Reviews**

We were very encouraged that the initial reviews established a strong and positive consensus from the outset. **All three reviewers initially positively rated our paper a 6**. This was further supported by a consistent positive assessment across all core criteria from every single reviewer, who each provided the following scores:

> Soundness: 3: good, Presentation: 3: good, Contribution: 3: good

They praised our work for addressing a **"novel problem"** (QWGa) of significant **"importance"** (S8oG), with one noting its relevance as a **"core task that many industrial actors are currently trying to tackle"** (QWGa). They found our empirical contribution to be a key strength, describing the experiments as **"thorough"** (m2c3), the results as **"strong"** (QWGa), and our method as **"meaningfully better than previous methods"** (m2c3). Finally, this strong technical contribution was matched by the quality of the manuscript itself, which was universally commended for its clarity, with one reviewer noting its **"perfect writing quality"** (QWGa) and another its **"clear logic flow"** (m2c3).

**Summary of Rebuttal and Revisions**

Building on a strong initial consensus from the reviewers, we addressed every concern and question with detailed explanations and significant manuscript revisions, including three new sets of experiments. The following is an itemized breakdown of this process:

**A. High-Level Framing and Contribution**

*   **Concern (m2c3):** The method's scope might be limited to Jericho, making it appear over-engineered.

    *   **Our Answer:** We explained that the core framework, which evolves an agent's `(prompt, memory, tools)` configuration, is domain-agnostic.

    *   **Our Revision:** **Added a discussion to the Introduction** with concrete examples of applying EvoTest to web navigation and coding agents.

*   **Concern (QWGa):** The paper felt more like an "engineering paper" than a fundamental scientific contribution for ICLR.

    *   **Our Answer:** We positioned the work as proposing a new paradigm for Test-Time Learning (TTL) and as a principled, data-efficient alternative to online RL.

    *   **Our Revision:** **Revised the Introduction** to sharpen the scientific framing around TTL and the concept of "semantic credit assignment."

---

> ### Author Response · Authors · 2025-12-02
>
> **B. Empirical Rigor and New Experiments**
>
> *   **Concern (QWGa):** The initial submission lacked second-order metrics (mean +/- std).
>
>     *   **Our Answer:** We acknowledged the point's importance and allocated resources to address it.
>
>     *   **Our Revision:** **Reran all main experiments over 5 seeds and added a new table to Appendix F** reporting full mean and standard deviation for all methods.
>
> *   **Concern (S8oG):** The comparison with fine-tuning methods was unfair because EvoTest used a proprietary LLM.
>
>     *   **Our Answer:** We agreed a direct comparison was essential for a fair claim.
>
>     *   **Our Revision:** **Ran a new experiment where EvoTest uses the exact same open-source model (qwen3-32b)** as the baselines. **Added these results to Section 5.3**, confirming that our framework's advantage holds even in a strictly fair comparison.
>
> *   **Concern (m2c3):** It was unclear how much performance gain came from our Evolver logic versus a simpler prompt optimization method.
>
>     *   **Our Answer:** We explained the conceptual difference between our method and generic textual mutations.
>
>     *   **Our Revision:** **Conducted a new ablation study ("EvoTest w/ Simple Mutation")** to isolate this effect and **added the results to Section 5.3**, quantifying the significant contribution of our structured evolutionary logic.
>
>
> **C. Methodological Clarity and Assumptions**
>
> *   **Concern (QWGa, S8oG):** The paper was missing a Limitations section.
>
>     *   **Our Answer:** We agreed this was a critical omission.
>
>     *   **Our Revision:** **Added a comprehensive "Limitations" section to the appendix**, discussing the trade-offs of our gradient-free approach and key dependencies.
>
> *   **Concern (S8oG):** We made a "hidden assumption" about immediate credit assignment for the success memory.
>
>     *   **Our Answer:** We clarified this is a simple heuristic for one part of the system, while the more complex credit assignment happens in the prompt evolution.
>
>     *   **Our Revision:** **Added a discussion** clarifying this distinction and explaining how the framework could be adapted for tasks with only final-episode rewards.
>
> *   **Concern (S8oG):** A claim that the Evolver could learn to change hyperparameters was unsubstantiated.
>
>     *   **Our Answer:** We agreed to provide empirical evidence.
>
>     *   **Our Revision:** **Added a new plot to the appendix** showing the evolution of the LLM decoding temperature over an episode, visually demonstrating this adaptive behavior.
>
> *   **Concern (m2c3, S8oG):** The UCB selection process was unclear.
>
>     *   **Our Answer:** We clarified that a single configuration is chosen for an entire episode, and evaluation is per-episode, not per-step.
>
>     *   **Our Revision:** **Revised Section 4.3** to state the per-episode nature of the selection-execution-evaluation loop.
>
> *   **Concern (S8oG):** The rationale for choosing specific LLM backbones was not explained.
>
>     *   **Our Answer:** We explained our reasoning based on cost, capability, and the ability to fine-tune.
>
>     *   **Our Revision:** **Added a paragraph to Section 5.1** making this rationale clear.
>
>
> **D. Presentation and Minor Issues**
>
> *   **Concern (S8oG):** The rich appendix was not well-referenced in the main text.
>
>     *   **Our Answer:** We committed to fixing this for readability.
>
>     *   **Our Revision:** **Performed a thorough pass on the manuscript to add direct references** to the appendices in all relevant sections.
>
> *   **Concern (S8oG):** The figures had small fonts and repetitive legends.
>
>     *   **Our Answer:** We committed to fixing this for readability.
>
>     *   **Our Revision:** **Revised all figures**, increasing font sizes and consolidating legends as promised.
>
> *   **Concern (m2c3, S8oG):** Minor typos (Equation 5, quotation marks, stray words).
>
>     *   **Our Answer:** We committed to fixing this for readability.
>
>     *   **Our Revision:** **Corrected all identified typos** and performed a full proofread of the manuscript.
>
>
> As this detailed summary shows, we took the review process very seriously and worked diligently to improve our paper. We were grateful to receive a follow-up comment from Reviewer S8oG, who wrote, **"I thank the authors for addressing my concerns and improving the quality of the submission. I increased my score."** Due to the timing of the OpenReview incident, we did not receive further follow-up from the other reviewers.
>
> **In summary, our paper began with a consistent "accept" recommendation, and our diligent rebuttal further improved the manuscript, leading to a score increase from the only reviewer who reply.**
>
> Thank you again for your time and your vital service to the community.
>
> Sincerely,
>
> The Authors of Submission 7583

---

### Meta-Review · Area_Chair_EfmA · 2026-01-06

**Summary:**

AI agents struggle with test-time learning in novel environments. To address this, the paper introduces the Jericho Test-Time Learning (J-TTL) benchmark for measuring and driving progress. The paper then proposes EvoTest, an evolutionary test-time learning framework in which an "Evolver Agent" revises the entire agent's configuration (prompt, memory, and tools) after each episode, leading to improved performance. Overall, the reviewers agree on the score, leaning towards accepting the paper. One reviewer points out that the paper makes some claims without providing supporting evidence, omits details here and there, and does not clearly explain all of its choices, making it difficult for others to properly evaluate or reproduce the work. I do agree on this, in particular, a limitation section should be provided. Another reviewer argues that it reads a bit like an engineering paper. I am not sure I fully agree, as Table 1, e.g., is showing how existing LLMs fail. However, I do agree that, for example, more motivational examples could have been provided and Section 4.1 leaves many details open, and these details can only be found in the appendix. This is not well-balanced. In particular, I am missing an ablation study that also considers the robustness of the prompt and co as well as an OOD testing when the knowledge required to solve a problem is not covered by the LLM. While the reviewers are more positive, these concerns need to be addressed to show how difficult AI still is. It is also not clear why only a subset of 6 games is considered. Should future comparison also use those? Should one explore other subset, but risk that one is not comparable? Should the 6 be called J-TTL-core and be placed as main benchmark?)

**Reviewer Concerns:**

The rebuttal addressed many concerns but did not fully resolve all of them. The authors provided a detailed ablation of the prompt evaluation and additional details that can be found in the appendix. However, some issues remain open. For example, the question of game selection needs further discussion; testing on additional subsets would be valuable but requires more time. The concerns about out-of-distribution performance and prompt robustness were not completely addressed. Overall, the major concerns were tackled, though some points would have benefited from additional rebuttal rounds.

**Reviewer Scores:**

The reviewers would have likely improved their scores based on the rebuttal, as many of their issues raised were addressed through the additional details and ablation studies provided by the authors.

---

### Decision · Program_Chairs · 2026-01-26

Accept (Poster)